



# Fast Infrared Radiative Transfer Calculations Using Graphics Processing Units: JURASSIC-GPU v2.0

Paul F. Baumeister[1] and Lars Hoffmann[1]

[1]Jülich Supercomputing Centre, Forschungszentrum Jülich, Jülich, Germany

**Correspondence:** Paul Baumeister (p.baumeister@fz-juelich.de)

**Abstract.** Remote sensing observations in the mid-infrared spectral region ($4 - 15\,\mu$m) play a key role in monitoring the composition of the Earth's atmosphere. Mid-infrared spectral measurements from satellite, aircraft, balloon and ground-based instruments provide information on pressure and temperature, trace gases as well as aerosols and clouds. As state-of-the-art instruments deliver a vast amount of data on a global scale, their analysis, however, may require advanced methods and high-performance computing capacities for data processing. A large amount of computing time is usually spent on evaluating the radiative transfer equation. Line-by-line calculations of infrared radiative transfer are considered to be most accurate, but they are also most time-consuming. Here, we discuss the emissivity growth approximation (EGA), which can accelerate infrared radiative transfer calculations by several orders of magnitude compared with line-by-line calculations. As future satellite missions will likely depend on Exascale computing systems to process their observational data in due time, we think that the utilization of graphical processing units (GPUs) for the radiative transfer calculations and satellite retrievals is a logical next step in further accelerating and improving the efficiency of data processing. Focusing on the EGA method, we first discuss the implementation of infrared radiative transfer calculations on GPU-based computing systems in detail. Second, we discuss distinct features of our implementation of the EGA method, in particular regarding the memory needs, performance, and scalability on state-of-the-art GPU systems. As we found our implementation to perform about an order of magnitude more energy-efficient on GPU-accelerated architectures compared to CPU, we conclude that our approach provides various future opportunities for this high-throughput problem.



# 1 Introduction

Mid-infrared radiative transfer, covering the spectral range of about $4$ and $15\,\mu m$ of wavelength, is of fundamental importance for various fields of atmospheric research and climate science. In the longwave part of the electromagnetic spectrum, the Earth's energy budget is balanced by thermally radiating back to space the energy received via shortwave solar radiation in the ultraviolet, visible, and near infrared spectral regions (Trenberth et al., 2009; Wild et al., 2013). Furthermore, a wealth of trace gases has distinct rotational-vibrational emission wavebands in the mid-infrared spectral region. Infrared spectral measurements are therefore utilized by many remote sensing instruments to measure atmospheric state parameters, like pressure, temperature, the concentrations of water vapor, ozone, various other trace gases, as well as cloud and aerosol particles (Thies and Bendix, 2011; Yang et al., 2013; Menzel et al., 2018).

Today, mid-infrared radiance measurements provided by satellite instruments are often assimilated directly for global forecasting, climate reanalyses, or air quality monitoring by national and international weather services and research centers. For example, observations by the fleet of Infrared Atmospheric Sounding Interferometer (IASI) instruments (Blumstein et al., 2004; Clerbaux et al., 2009) from the European polar orbiting MetOp satellites have become a backbone in numerical weather prediction next to monitoring atmospheric composition. New space-borne missions utilizing the wealth of information from hyperspectral mid-infrared observations will be launched beyond 2020, e. g., the InfraRed Sounder (IRS) instrument on the Meteosat Third Generation (MTG) geostationary satellite for near real-time weather forecasting systems and the European Space Agency's new FORUM mission (ESA, 2019).

The vast amount of remote sensing observations from next-generation satellite sensors poses a big data challenge for Earth system scientists. Fast and accurate radiative transfer models (RTMs) for the Earth's atmosphere are a key component for the analysis of these observations. However, many traditional RTMs suffer from large computational costs, typically requiring high performance computing resources to process multi-year satellite missions. In this study, we focus on the porting and optimization of radiative transfer calculations for the mid-infrared spectral region to graphics processing units (GPUs).

GPUs bear a high potential for accelerating atmospheric radiative transfer calculations. For instance, Mielikainen et al. (2011) developed a fast GPU-based radiative transfer model for the Infrared Atmospheric Sounder (IASI) instruments aboard the European MetOp satellites. The model of Mielikainen et al. (2011) estimates band transmittances for individual IASI channels with a regression approach (McMillin and Fleming, 1976; Fleming and McMillin, 1977; McMillin et al., 1979). It was developed to run on a low-cost personal supercomputer with 4 NVIDIA Tesla C1060 GPUs with a total of 960 cores, delivering near $4\,\mathrm{TFlop/s}$ theoretical peak performance. On that system, the study found that 3600 full IASI spectra (8461 channels) can be calculated per second. More recently, Mielikainen et al. (2016) examined the feasibility of using GPUs to accelerate the rapid radiative transfer model (RRTM) shortwave module for massive amounts of atmospheric profiles. The study of Mielikainen et al. (2016) found that NVIDIA's Tesla K40 GPU card can provide a speedup of more than 200 compared to its single-threaded Fortran counterpart running on Intel Xeon E5-2603 CPU.

An alternative approach to perform atmospheric infrared radiative transfer calculations on field-programmable gate arrays (FPGAs) was proposed by Kohlert and Schreier (2011). Starting point for their study were radiative transfer calculations based



on the line-by-line approach. In this approach the transmittance is determined by evaluating the temperature- and pressure-dependent line intensities and line shapes of all relevant molecular absorption lines. However, in the infrared, this may require to evaluate hundreds to thousands of absorption lines in a given spectral region. Kohlert and Schreier (2011) ported this most intensive part of the line-by-line calculations to FPGAs. Performance tests showed that the computation time for 950 absorption

lines on 16000 spectral grid points was about 0.46 s on their test system. This is several orders of magnitude slower than both the regression and the EGA approach, but it needs to be kept in mind that the line-by-line methods generally provides the most accurate results.

In this study, we will discuss porting and performance analyses of radiative transfer calculations based on the emissivity growth approximation (EGA) method as implemented in the Juelich Rapid Spectral Simulation Code (JURASSIC) to GPUs.

The EGA method as introduced by Weinreb and Neuendorffer (1973), Gordley and Russel (1981) and Marshall et al. (1994) has been successfully applied for the analysis of various satellite experiments over the past 30 years. This includes, for instance, the Cryogenic Limb Array Etalon Spectrometer (CLAES) (Gille et al., 1996), the Cryogenic Infrared Spectrometers and Telescopes for the Atmosphere (CRISTA) (Offermann et al., 1999; Riese et al., 1999), the Halogen Occultation Experiment (HALOE) (Gordley et al., 1996), the High Resolution Dynamics Limb Sounder (HIRDLS) (Francis et al., 2006), the Sounding of the

Atmosphere using Broadband Emission Radiometry (SABER) (Mertens et al., 2001; Remsberg et al., 2008), the Stratospheric Aerosol and Gas Experiment (SAGE-III) (Chiou et al., 2003), and the Solar Occultation for Ice Experiment (SOFIE) (Rong et al., 2010) satellite instruments.

The JURASSIC model was first described by Hoffmann et al. (2005) and Hoffmann et al. (2008), discussing the retrieval of chlorofluorocarbon concentrations from Michselson Interferometer for Passive Atmospheric Sounding (MIPAS) observations

aboard ESA's Envisat. Hoffmann et al. (2009), Weigel et al. (2010), and Ungermann et al. (2012) applied JURASSIC for trace gas retrievals from infrared limb observations of the Cryogenic Infrared Spectrometers and Telescopes - New Frontiers (CRISTA-NF) aircraft instrument. Hoffmann and Alexander (2009), Meyer and Hoffmann (2014), and Meyer et al. (2018) extended JURASSIC for nadir observations and applied the model for retrievals of stratospheric temperature from Atmospheric InfraRed Sounder (AIRS) observations aboard NASA's Aqua satellite. Hoffmann et al. (2014) and Hoffmann et al. (2016) used

the model to analyze AIRS observations of volcanic aerosols. Preusse et al. (2009) and Ungermann et al. (2010a) applied JURASSIC for tomographic retrievals of stratospheric temperature from high-resolution limb observations. Ungermann et al. (2010b) and Ungermann et al. (2011) extended JURASSIC for tomographic retrievals of high-resolution aircraft observations of the Global Limb Imager of the Atmosphere (GLORIA) instrument. Griessbach et al. (2013, 2014, 2016) extended JURASSIC to allow for simulations of scattering of infrared radiation on aerosol and cloud particles.

The GPU-enabled version of the JURASSIC radiative transfer model described here is referred to as JURASSIC-GPU. The first version of JURASSIC-GPU was developed and introduced by Baumeister et al. (2017). This paper introduces version 2.0 of the JURASSIC-GPU model, which has been significantly revised and optimized for more recent GPU cards, namely NVIDIA's A100 Tensor-Core GPUs as utilized in the Jülich Wizard for European Leadership Science (JUWELS) booster module at the Jülich Supercomputing Centre (JSC), Germany. Whereas version 1.0 of JURASSIC-GPU was considered to be





a proof of concept, we consider the version 2.0 of JURASSIC-GPU described here to be ready for production runs on the
JUWELS booster module.

In Sect. 2, we provide a comprehensive description of the algorithms implemented in the JURASSIC radiative transfer
model, which has not been presented in the literature so far. In particular, we describe the ray tracing algorithm, the differences
between monochromatic radiative transfer and the band-transmittance approximation, the use of emissivity look-up tables, the
emissivity growth approximation (EGA), and the radiance integration along the line-of-sight. In Sect. 3, we discuss the porting
of JURASSIC to GPUs, including a summary of the algorithmic changes and improvements, as well as detailed performance
analyses with comparisons between CPU and GPU calculations using the JURASSIC-GPU model. Section 4 provides the
summary, conclusions, and an outlook.

## 2  JURASSIC model description

### 95  2.1  Definition of atmospheric state and ray tracing

The first step in numerical modeling of infrared radiative transfer is the definition of the atmospheric state. In JURASSIC,
the atmosphere is assumed to be homogeneously stratified and field quantities such as pressure $p_i$, temperature $T_i$, volume
mixing ratios $q_{j,i}$ (with trace gas index $j$), and aerosol extinction coefficients $k_{l,i}$ (with spectral window index $l$) are specified
in the form of vertical profiles on levels $i = 1, \ldots, n$. Linear interpolation is applied to $\log(p_i)$, $T_i$, $q_{j,i}$, and $k_{l,i}$ to determine
the atmospheric state in between the given height levels. For coordinate transformations between spherical and Cartesian
coordinates, JURASSIC assumes the Earth to be spherical with a fixed mean radius $R_e$ of 6367.421 km.

Once the atmospheric state is defined, the ray paths through the atmosphere need be calculated. Here it needs to be considered
that refraction in the Earth's atmosphere leads to bending of the ray paths towards the Earth's surface. In case of the limb
sounding geometry, this effect causes real tangent heights in the troposphere to be lowered up to several hundred meters below
geometrically calculated tangent heights that have been calculated without refraction being considered.

The positions along a single ray path $\mathbf{r}(s)$ are calculated numerically by means of the Eikonal equation (Born and Wolf,
1999),

$$\frac{d}{ds}\left(n\frac{d\mathbf{r}}{ds}\right) = \nabla n. \tag{1}$$

Here, $s$ is the spatial coordinate along the ray path with the origin $s = 0$ being located at the position of the observing instru-
ment. The refractive index $n$ depends on wavelength $\lambda$, pressure $p$, temperature $T$ and water vapour partial pressure $e$ (Ciddor,
1996),

$$n = 1 + \left(N_g \frac{p}{p_0}\frac{T_0}{T} - \frac{e}{T}\frac{11.27\,\mathrm{K}}{\mathrm{hPa}}\right) \times 10^{-6}. \tag{2}$$

In Eq. (2), $N_g$ indicates the refractivity of dry air for standard conditions ($p_0 = 1013.25\,\mathrm{hPa}$ and $T_0 = 273.15\,\mathrm{K}$),

$$N_g = 287.6155 + 4.8866\frac{\mu\mathrm{m}^2}{\lambda^2} + 0.068\frac{\mu\mathrm{m}^4}{\lambda^4}. \tag{3}$$





In the mid-infrared spectral region $(4\ldots15\,\mu\text{m})$, the variations with wavelength are typically negligible (deviations in $N_g$ with respect to $\lambda$ are less than 0.1 %). A comparison of the terms in Eq. (2) for different climatological conditions showed that water vapor in this wavelength range also has no substantial influence on refraction (variations in $n-1$ smaller 0.5 %). For these reasons, JURASSIC applies a simplified equation to calculate the refractivity,

$$n \approx 1 + 7.753 \times 10^{-5}\frac{p}{T}\frac{\text{K}}{\text{hPa}}. \tag{4}$$

In JURASSIC, the Eikonal equation (1) is solved numerically in Cartesian coordinates by means of an iterative scheme described by Hase and Höpfner (1999),

$$\mathbf{r}_{i+1} = \mathbf{r}_i + 0.5\,ds\,(\mathbf{e}_{t,i} + \mathbf{e}_{t,i+1}), \tag{5}$$

$$\mathbf{e}_{t,i+1} = \frac{\mathbf{e}_{t,i}\,n(\mathbf{r}_i) + ds\,\nabla n(\mathbf{r}_i + 0.5\,ds\,\mathbf{e}_{t,i})}{|\mathbf{e}_{t,i}\,n(\mathbf{r}_i) + ds\,\nabla n(\mathbf{r}_i + 0.5\,ds\,\mathbf{e}_{t,i})|}. \tag{6}$$

The determination of the ray path starts at the location $\mathbf{r}_0$ of the instrument. By specifying a second position in the atmosphere, referred to as the view point, the initial tangent vector $\mathbf{e}_{t,0}$ is defined. For instance, in the limb geometry, the geometrical tangent point can be selected as the view point, but this is not mandatory. With this rather general approach for ray-tracing, any observation geometry (limb, nadir, zenith, or occultation) for instruments located inside or outside of the atmosphere can be defined.

The step size $ds$ along the ray path is the most important control parameter regarding the speed and accuracy of the radiative transfer calculations. Over a large range of choices of $ds$, the mean computation time $t$ for the determination of the ray paths and the subsequent calculation of the radiative transfer is proportional to the reciprocal of the step size, $t \sim 1/ds$. If the step size is selected too small, the calculation of the radiative transfer takes too much computing time. At larger step sizes, the positions of the points along the ray path may still be determined quite well, but the inhomogeneity of the atmosphere along the ray paths is only insufficiently sampled. The extent of these errors depends on the individual atmospheric conditions. In JURASSIC, the default maximum step size along a ray path is 10 km, which is suitable for the limb geometry. To make the method suitable also for the nadir geometry, an additional constraint is imposed, which will reduce individual step sizes to ensure that the vertical component of the steps will not become larger than 500 m by default.

For the limb sounding geometry, it is of particular interest to know the actual real tangent height of the ray paths when taking into account refraction. For this purpose, JURASSIC applies a parabolic interpolation based on the three points of the ray path closest to Earth surface to enable a more accurate determination of the tangent point. The error of the tangent heights estimated by the parabolic interpolation method was found to be $1-2$ orders of magnitude below the accuracy by which this quantity can typically be measured.





## 2.2 Monochromatic radiative transfer

The propagation of monochromatic radiance $I$ along a ray path through the atmosphere is calculated from the radiative transfer
equation (Chandrasekhar, 1960)

$$I(\nu, x) = I(\nu, 0)\tau(\nu, 0, x) + \int_0^x J(\nu, x')\frac{d}{dx'}\tau(\nu, x', x)\,dx'. \tag{7}$$

Here, $\nu$ denotes the wavenumber, $x$ is the position along the ray path, $I(\nu, 0)$ is the radiation entering the ray path at the starting point, $I(\nu, x)$ is the radiation exiting at the other end, $\tau(\nu, x', x)$ is the transmission along the path from $x'$ to $x$ and $J(\nu, x')$ is the source function, which describes both, the thermal emissions of the emitters along the path and the scattering of radiation
into the path. Note that at this point we changed from the local coordinate $s$ used for ray-tracing to the local coordinate $x$ for the radiative transfer calculations, which is running in the opposite direction.

In case of local thermodynamic equilibrium and if scattering of radiation can be neglected, the source function corresponds to the Planck function,

$$B(\nu, T) = \frac{2hc^2\nu^3}{\exp[hc\nu/(kT)] - 1}, \tag{8}$$

with Planck constant $h$, speed of light $c$ and Boltzmann constant $k$. Deviations from the local thermodynamic equilibrium usually occur only above the stratopause (López-Puertas and Taylor, 2002). Scattering effects can be neglected in the mid-infrared spectral range for clear air conditions, meaning that any significant concentrations of clouds or aerosol particles are being absent (Höpfner and Emde, 2005; Griessbach et al., 2013).

The transmissivity of the atmosphere is determined by specific molecular rotational-vibrational wavebands of the trace gases
and by a series of continuum processes. For molecular emitters, the transmissivity along the ray path is related to the absorption coefficients $\kappa_i$ and particle densities $\rho_i$ according to

$$\tau(\nu, x', x) = \exp\left[-\int_{x'}^x \sum_i \kappa_i(\nu, x'')\,\rho_i(x'')\,dx''\right], \tag{9}$$

where $i$ refers to the trace gas index. The absorption coefficients can be calculated by summation over a large number of spectral lines,

$$\kappa_i(\nu, x'') = \sum_j k_{i,j}[T(x'')]\,f_{i,j}[\nu, p(x''), T(x'')], \tag{10}$$

where $j$ is the line index. The parameters for determining the line intensity $k_{i,j}$ and shape function $f_{i,j}$ for a given pressure $p$ and temperature $T$ along the line-of-sight can be obtained from spectroscopic databases. The HITRAN (High Resolution Transmission) database (Rothman et al., 2009, 2013) is used in the present work. The HITRAN database contains the parameters for millions of spectral lines of more than 40 trace gases as well as directly measured infrared absorption cross sections of
more complex molecules (e. g., chlorofluorocarbons).





Some species are significantly affected by line mixing, where the interaction or overlap of spectral lines can no longer be described by simple addition (Strow, 1988). Line mixing is taken into account for $CO_2$ in our calculations, but will not be further discussed here in detail. Furthermore, for some trace gases at comparatively high pressure in the troposphere and lower stratosphere, especially for $CO_2$, $H_2O$, $N_2$, and $O_2$, the individual spectral lines merge and form a continuum-like emission background (Lafferty et al., 1996; Thibault et al., 1997; Mlawer et al., 2012). Continuum emissions are also caused by aerosol and cloud particles. In JURASSIC, the influence of aerosols and clouds on the transmission can be described by direct specification of extinction coefficients $k$. The extinction coefficients indicate the attenuation of radiance per path length and are linked to the transmission according to

$$\tau(\nu, x', x) = \exp\left[-\int_{x'}^{x} k(\nu, x'')\, dx''\right]. \tag{11}$$

## 2.3 Band transmittance approximation

Satellite instruments measure radiance spectra at a given spectral resolution. The mean radiance $\bar{I}$ measured by an instrument in a given spectral channel is obtained by spectral integration of the monochromatic radiance spectrum,

$$\bar{I} = \int_{\nu_0}^{\nu_1} f(\nu) \int_{0}^{\infty} B[\nu, T(s)] \frac{d}{ds} \varepsilon(\nu, s, 0)\, ds\, d\nu, \tag{12}$$

The filter function $f(\nu)$ indicates the normalized spectral response with respect to wavenumber $\nu$ for the instrument channel being considered.

For simplicity, we focus here on the case of limb sounding, where the origin of the ray paths is in cold space and the source term $I(s \to \infty) \approx 0$ can be omitted. Furthermore, we replaced the transmission $\tau$ by the emissivity $\varepsilon$,

$$\varepsilon(\nu, s, 0) = 1 - \tau(\nu, s, 0). \tag{13}$$

which is more suitable for numerical evaluation considering possible loss of accuracy in the representation of $\tau$ compared to $\varepsilon$ under optically thin conditions.

The most accurate method for calculating the monochromatic emissivity $\varepsilon$ is the line-by-line evaluation as discussed in Sect. 2.2. However, the line-by-line method is computationally most demanding because, depending on the spectral range, tens of thousands of molecular emission lines may need to be considered. For each line, the line intensity and shape function must be determined depending on pressure and temperature. One option to accelerate the radiative transfer calculations is an approximate solution described by Gordley and Russel (1981) and Marshall et al. (1994). A slightly modified formulation of





Rodgers (2000) has been used here,

$$\bar{I} \approx \int\limits_0^\infty \bar{B}[T(s)] \frac{d}{ds} \bar{\varepsilon}(s,0) \, ds, \tag{14}$$

$$\bar{\varepsilon}(s,0) = \int\limits_{\nu_0}^{\nu_1} f(\nu) \varepsilon(\nu,s,0) \, d\nu, \tag{15}$$

$$\bar{B}(T) = \int\limits_{\nu_0}^{\nu_1} f(\nu) B(\nu,T) \, d\nu. \tag{16}$$

Instead of detailed monochromatic calculations of the radiance, emissivity, and Planck function, this approximation uses only the spectral averages (indicated by bars) within the spectral range defined by the filter function. Accordingly, the method is referred to as the band transmittance approximation. This method can become computationally very fast, if the spectrally averaged emissivities are determined by means of a band model or, as is the case for JURASSIC, by using a set of look-up tables that have been precalculated by means of a line-by-line model.

Equations (14) to (16) provide only an approximate solution of the radiation transfer, as the spectral correlations of the emissivity and its derivative along the ray path with the Planck function are neglected. The corresponding error is given by

$$\Delta I = \int\limits_0^\infty \int\limits_{\nu_0}^{\nu_1} f(\nu) [B(\nu,T) - \bar{B}(T)] \frac{d}{ds} ds [\varepsilon(\nu,s,0) - \bar{\varepsilon}(s,0)] \, d\nu \, ds. \tag{17}$$

The error due to neglecting these correlations can be significantly reduced by using more sophisticated methods for determining the spectral mean emissivities along the path, in particular by means of the emissivity growth approximation (EGA).

Another error of the band transmittance approximation results from the fact that the spectral correlations of different emitters are neglected. From Eq. (9) it can be seen that the monochromatic total transmission of several emitters results multiplicatively from their individual transmissions. As the total monochromatic emissivity is calculated from

$$\varepsilon(\nu) = 1 - \tau(\nu) = 1 - \prod_i \tau_i(\nu) = 1 - \prod_i [1 - \varepsilon_i(\nu)], \tag{18}$$

the spectral mean emissivity is given by

$$\bar{\varepsilon} = 1 - \prod_i (1 - \bar{\varepsilon}_i) + \Delta\varepsilon. \tag{19}$$

The residual $\Delta\varepsilon$ comprises the spectral correlations between all the emitters. For example, for a pair of two emitters, the residual is given by

$$\Delta\varepsilon = \int\limits_{\nu_0}^{\nu_1} f(\nu) [\varepsilon_1(\nu) - \bar{\varepsilon}_1][\varepsilon_2(\nu) - \bar{\varepsilon}_2] \, d\nu. \tag{20}$$

Such correlation terms are neglected in the JURASSIC model. The associated errors remain small, if at least one of the emitters

has a relatively constant spectral response. The approximation $\Delta\varepsilon \approx 0$ is therefore referred to as the continuum approximation.





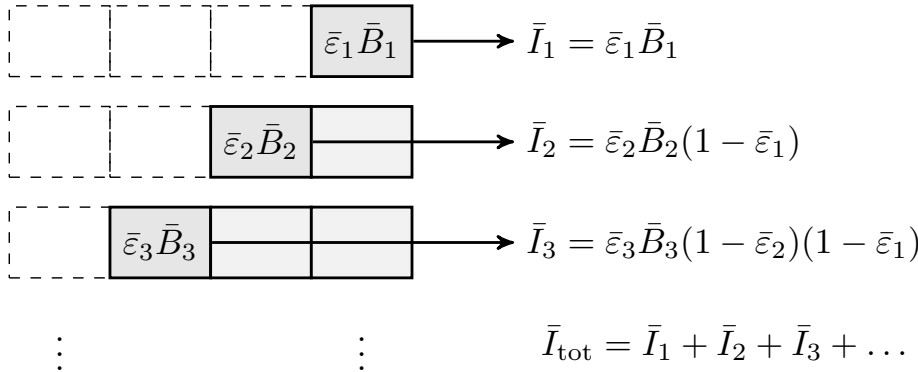

**Figure 1.** Integration of radiance along a ray path. Contributions $\bar{\varepsilon}_i \bar{B}_i$ to the total radiance $\bar{I}_{\text{tot}}$ originate in every segment $i$ of the ray path through the atmosphere. The radiance contributions are partly absorbed due to the transmittance $\bar{\tau}_{1,i-1} = (1-\bar{\varepsilon}_1) \times (1-\bar{\varepsilon}_2) \times \cdots \times (1-\bar{\varepsilon}_{i-1})$ along the path. The observer is located on the right side in this sketch.

In practice, the errors of the continuum approximation also remain small, if a sufficiently large spectral range with many spectral lines is considered. Nevertheless, it is recommended that the errors associated with the band transmittance and continuum approximations are assessed by means of comparisons with line-by-line calculations.

### 2.4 Numerical integration along the line-of-sight

For the numerical integration of the approximated radiative transfer equation, Eq. (14), the ray path is first divided into segments. For each segment, homogeneous atmospheric conditions are assumed. The spectral mean emissivity and the value of the Planck function of each segment are calculated from pressure, temperature, trace gas volume mixing ratios, and aerosol extinction coefficient data provided by the raytracing algorithm and by applying the trapezoidal rule for sampling. The approximated radiative transfer equation, Eq. (14), is then solved by an iterative scheme,

$$I_{k+1} = I_k + \bar{\varepsilon}_{k+1}\bar{B}_{k+1}\bar{\tau}_k^*, \tag{21}$$

$$\bar{\tau}_k^* = (1-\bar{\varepsilon}_k)\bar{\tau}_{k-1}^*, \tag{22}$$

$$\tau_0^* = 1, \quad I_0 = 0. \tag{23}$$

The principle of this scheme is illustrated in Fig. 1. Assuming that the ray path already consists of $k$ segments that produce the radiance $I_k$ at the location of the instrument, adding a further segment with index $k+1$, an additional radiance contribution

$\bar{\varepsilon}_{k+1}\bar{B}_{k+1}$ will incident on the path of $k$ segments. On the way to the instrument, the incident radiation is partially absorbed, whereby the absorption is determined by the total transmission $\bar{\tau}_k^*$ of the $k$ segments along the path. By adding the $(k+1)$-th segment, the instrument receives an additional radiance contribution $\bar{\varepsilon}_{k+1}\bar{B}_{k+1}\bar{\tau}_k^*$. The path transmission $\bar{\tau}_k^*$ is calculated by multiplying the transmissions $\bar{\tau}_k = 1 - \bar{\varepsilon}_k$ of the individual segments.





## 2.5 Use of emissivity look-up tables

The full advantage in terms of speed of the approximated radiative transfer calculations can be obtained by using look-up tables of spectrally averaged emissivities, which have been prepared for JURASSIC by means of line-by-line calculations. In subsequent radiative transfer calculations, the spectral emissivities are determined by means of simple and fast interpolation from the look-up tables. For the calculation of the emissivity look-up tables, any conventional radiative transfer model can be used, which allows to calculated the transmission of a homogeneous gas cell depending on pressure $p$, temperature $T$,

and emitter column density $u = q\,p/(kT)\,ds$. For the calculations shown here, the MIPAS Reference Forward Model (RFM) (Dudhia, 2017) has been used to generate the emissivity look-up tables.

The pressure, temperature, and column density values in the emissivity look-up tables need to cover the full range of atmospheric conditions. If the coverage or the sampling of the tables is too low, this could significantly worsen the accuracy of the radiative transfer calculations. We calculated the look-up tables for pressure levels $p_i = p_0 \exp(-z_i/7\,\mathrm{km})$ with

$z_i = 0, 2, 4, \ldots, 80\,\mathrm{km}$. For each pressure level, temperatures at $\delta T_{i,j} = -70, -65, \ldots, +70\,\mathrm{K}$ around the mid-latitude climatological mean have been selected. For each $(p_i, T_{i,j})$ combination, the column densities $u_{i,j,k}$ of the respective emitter are chosen so that the mean emissivity $\bar{\varepsilon}(p_i, T_{i,j}, u_{i,j,k})$ covers the range $[10^{-6}, 0.9999]$ and that an increase of $12.2\,\%$ occurs between the column density grid points.

Since even a single forward calculation for a remote sensing observation may require thousands of interpolations on the

emissivity look-up tables, this process must be implemented to be most efficient. Fast interpolation across the look-up tables is ensured by applying the bisection method described by Press et al. (2002) for identifying the interpolation nodes. Interpolation itself is linear in pressure, temperature, and column density. Other interpolation schemes (e. g., second order polynomials, cubic splines, single or double logarithmic interpolations) may better represent the actual shape of the emissivity curves with a smaller number of grid points, but were rejected after testing due to the significantly increased computational effort for inter-

polation. For example, the numerical effort to calculate the logarithm or exponential function values in the double logarithmic interpolation is much larger than the numerical effort for linear interpolation, which only requires a single division and a single multiplication. We found that the increased numerical effort of higher-order interpolation methods cannot be compensated for by the fact that fewer grid points are required to provide accurate representation of the emissivity look-up tables.

As an example, Fig. 2 shows the emissivity look-up table for the 15 micron carbon dioxide Q branch at $669\,\mathrm{cm}^{-1}$ as used for

some of the radiative transfer calculations in this study. The emissivity curves shown here have been calculated with the RFM line-by-line model. The monochromatic absorption spectra provided by the RFM for a fixed pressure $p$ and temperature $T$ have been convoluted with a boxcar spectral response function with a width of $1\,\mathrm{cm}^{-1}$ to obtain the spectral mean emissivities. In the weak line limit, for small column densities $u$, the spectral mean emissivity $\bar{\varepsilon}$ scales linearly with $u$. In the strong line limit, when the line centers get saturated, absorption is controlled by the line wings and $\bar{\varepsilon}$ scales with the square root of $u$. For large

$u$, the emissivity curves will completely saturate, $\bar{\varepsilon} \to 1$.



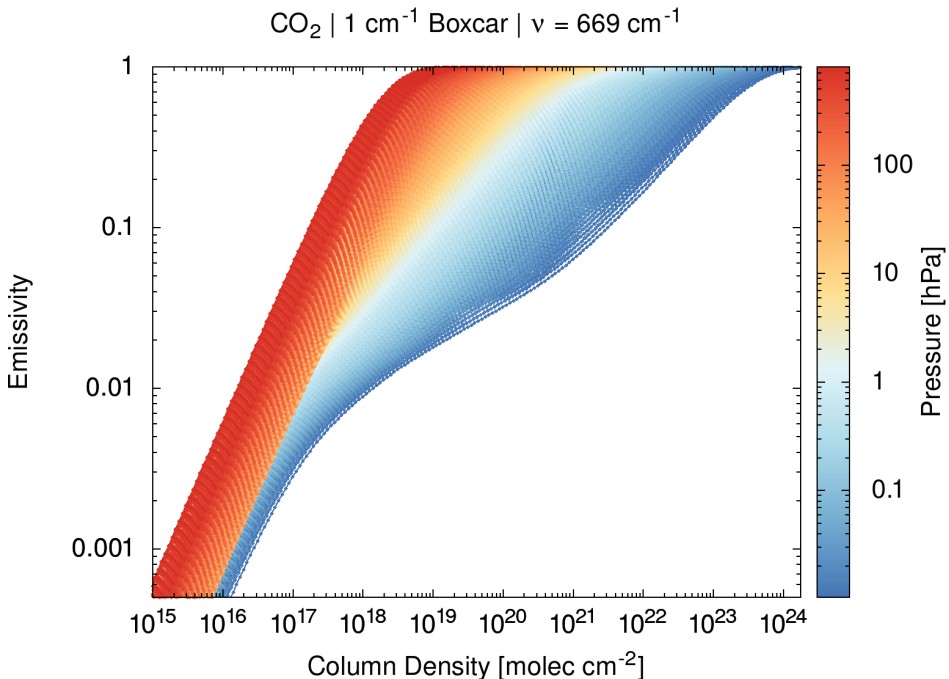

**Figure 2.** Log-log plot of spectral mean emissivity curves for carbon dioxide ($CO_2$) at $669\,\mathrm{cm}^{-1}$ and for a $1\,\mathrm{cm}^{-1}$ boxcar spectral response function. The curves have been calculated for different pressure levels (see color coding) and temperature values (not highlighted). Small dots along the emissivity curves illustrate the dense sampling of the look-up table data.

### 2.6 Emissivity growth approximation

Spectral mean emissivities of an inhomogeneous atmospheric path can be obtained from the look-up tables in different ways. In JURASSIC, the EGA method (Weinreb and Neuendorffer, 1973; Gordley and Russel, 1981) is applied. For simplicity, only the determination of emissivity in the case of a single emitter is described here. In the case of multiple emitters, the determination

proceeds analogously, followed by the determination of total emissivity according to Eq. (19). Within the EGA method, the spectral mean emissivity is interpolated from the look-up tables according to the following scheme:

1. For a given ray path of $k$ segments, to which a segment $k+1$ is to be added, we first determine an effective column density $u^*$ so that $\bar{\varepsilon}(u^*, p_{k+1}, T_{k+1}) = \bar{\varepsilon}_k^*$, where $\bar{\varepsilon}_k^*$ denotes the total emissivity of the path of $k$ segments. To determine $u^*$, an inverse interpolation needs to be performed on the emissivity look-up tables.

2. The emissivity of the extended path of $k+1$ segments is determined from $\bar{\varepsilon}_{k+1}^* = \bar{\varepsilon}(u^* + u_{k+1}, p_{k+1}, T_{k+1})$. Direct interpolation from the emissivity tables is applied in this case. The emissivity of the $(k+1)$-th segment is given by $\bar{\varepsilon}_{k+1} = 1 - (1 - \bar{\varepsilon}_{k+1}^*)/(1 - \bar{\varepsilon}_k^*) \approx \bar{\varepsilon}_{k+1}^* - \bar{\varepsilon}_k^*$.

The basic assumption of the EGA method is that the total emissivity of $k$ ray path segments can be transferred to the emissivity curve $\bar{\varepsilon}(u, p_{k+1}, T_{k+1})$ of the currently considered segment $k+1$ at pressure $p_{k+1}$ and temperature $T_{k+1}$ of the extended path,



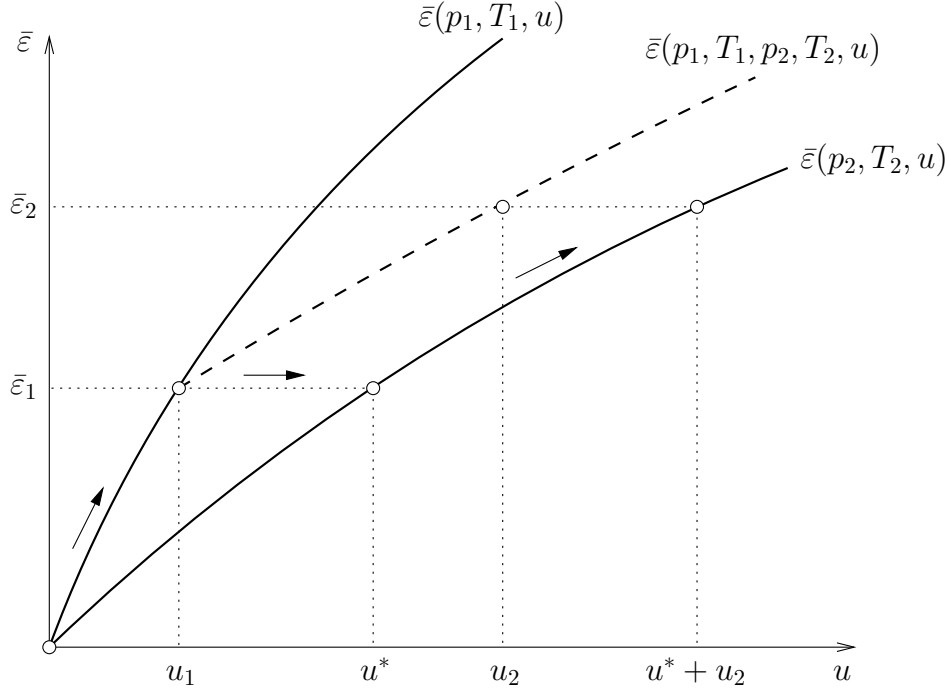

**Figure 3.** Illustration of the EGA method. For the first segment of a ray path, the EGA method will apply the emissivity curve $\bar\varepsilon(p_1, T_1, u)$ to determine the spectral mean segment emissivity. For the second segment, the EGA method will locate the pseudo column amount $u^*$ on $\bar\varepsilon(p_2, T_2, u)$ and calculate the emissivity of the extended path based on the total column amount $u^* + u_2$. This approach will be followed along the entire path.

and that the emissivity growth due to the additional column density $u_{k+1}$ of the segment $k+1$ can be calculated by following this emissivity curve, starting from the pseudo column amount $u^*$.

The principle of the EGA method is further illustrated in Fig. 3 for the first two segments of a ray path. To determine the emissivity $\bar\varepsilon_1$ of the first segment, we need to follow the emissivity curve $\bar\varepsilon(p_1, T_1, u)$ so that $\bar\varepsilon_1 = \bar\varepsilon(p_1, T_1, u_1)$. If a second segment is added, the real emissivity would follow an emissivity curve $\bar\varepsilon(p_1, T_1, p_2, T_2, u)$, so that $\bar\varepsilon_2 = \bar\varepsilon(p_1, T_1, p_2, T_2, u_2)$. 290 However, only an emissivity model for homogeneous paths is available from the look-up tables. Therefore the emissivity curve $\bar\varepsilon(p_2, T_2, u)$ is used to determine $\bar\varepsilon_2$. The difficulty is to determine the ideal starting point $u^*$ on $\bar\varepsilon(p_2, T_2, u)$. This starting point is characterized by the fact that the detailed spectral shape of the emissivity $\bar\varepsilon(p_1, T_1, u_1, \nu)$ corresponds to that at $\bar\varepsilon(p_2, T_2, u^*, \nu)$. The solution $\bar\varepsilon_2 = \bar\varepsilon(p_2, T_2, u^* + u_2)$ would then be exact. However, such a point usually does not exist. Within the framework of the EGA method, one assumes that the best starting point on $\bar\varepsilon(p_2, T_2, u)$ is the one where at least the spectrally averaged 295 emissivity is the same, i. e., $u^*$ is determined so that $\bar\varepsilon(p_2, T_2, u^*) = \bar\varepsilon_1$.





## 3 Infrared radiative transfer calculations on GPUs

### 3.1 Implementation details of JURASSIC-GPU

#### 3.1.1 Usage of the CUDA programming model

JURASSIC is written in the C programming language and makes use of only a few library dependencies. In particular, the GNU
Scientific Library (GSL) is used for linear algebra in the retrieval code provided along with JURASSIC. In order to connect the
GPU implementation of JURASSIC seamlessly to the reference implementation (Hoffmann, 2015), we restructured only those
parts of JURASSIC that perform intensive compute operations, i. e., the EGA method and also the ray tracer. While the EGA
calculations are typically most compute intensive, the ray tracing kernel has also been ported to the GPU mostly for reasons of
data locality so the ray path data can reside in GPU memory and does not require CPU-to-GPU data transfers before starting
the radiative transport calculation along the ray paths.

For the GPU programming, we selected the Compute Unified Device Architecture (CUDA) programming model, which
addresses exclusively NVIDIA graphical processors. CUDA is a dialect of C/C++ and the user has to write compute kernels in
a specific CUDA syntax:

```
      __global__ void clear_array_kernel(double a[]) {
310       a[blockIdx.x*blockDim.x + threadIdx.x] = 0;
      }

      __host__ int clear_array(double a[], size_t n) {
          <<< n/64, 64 >>> clear_array_kernel(a);
315       return n%64;
      }
```

This minimal example already shows some of the most important features of the CUDA programming model. There are kernels
with the attribute __global__ which can be executed on the GPU. Kernels need to be launched by regular CPU functions
which we call driver functions. All regular CPU functions inside the source files translated with the NVIDIA C compiler
need to be marked as __host__. The CUDA kernel launch consists of two parts. The left part in between triple chevrons
(<<< ... >>>) indicates the launch configuration, i. e., we specify the number of CUDA blocks and the number of CUDA
threads per block (here 64). The terminology might be misleading as CUDA threads are not like CPU threads (POSIX threads
or OpenMP threads). Rather, one should think about CUDA threads as lanes of a vector unit. We will use the term lanes here
instead of CUDA threads. CUDA blocks, however, exhibit some similarity to CPU threads.
Inside our example CUDA kernel clear_array_kernel, we access inbuilt variables: threadIdx.x and blockIdx.x
are the lane and block index, respectively, and blockDim.x will have the value 64 here due to the selected launch configura-
tion of 64 lanes per block.





The example so far assigns zero to the values of array `a[]` but behaves only correctly if `n` is a multiple of 64. In order to produce code that performs correctly for any launch configuration and without too complicated checks for corner cases, we make use of a technique called grid stride loops (Harris, 2013):

```
__global__ void clear_array_kernel(double a[], int n) {
    for(int i = blockIdx.x*blockDim.x + threadIdx.x; i < n;
            i += gridDim.x*blockDim.x)
        a[i] = 0;
}
```

The inbuilt variable `gridDim.x` will assume the value `n/64`. Now any launch configuration `<<< g, b >>>` with $g \geq 1$ and $1 \leq b \leq 1024$ will lead to correct execution. This is particularly helpful when tuning for performance as we can freely vary the number of lanes per block `b` and also the number of blocks `g`.

### 3.1.2 CPU versus GPU memory

Graphical processors are equipped with their own memory, typically based on a slightly different memory technology compared to standard CPU memory (SDRAM versus DRAM). Therefore, pointers to be dereferenced in a CUDA kernel need to reside in GPU memory, i. e., need to be allocated with `cudaMalloc`. The returned allocations reside in GPU memory and can only be accessed by the GPU. Before we can use the arrays on the CPU (e. g., for output) we have to manually copy-transfer these GPU memory regions into CPU memory of the same size. Likewise, data may have to be copied from CPU memory to GPU memory, before it can be used on the GPU. These data transfers between CPU and GPU memory are typically costly and should be avoided as far as possible.

In the unified memory model (supported by all modern NVIDIA GPUs) `cudaMallocManaged` returns allocations where memory transfers are hidden from the user, i. e., the coding complexity of having a CPU pointer and a GPU pointer and keeping the array content in sync is taken care of by the hardware. Our GPU implementation of JURASSIC uses unified memory for the look-up tables. For the observation geometry and atmosphere data (inputs) and the resulting radiances (results), JURASSIC makes use of GPU-only memory and performs explicit memory transfers to and from GPU-memory.

### 3.1.3 Single source code policy

For production codes, one might accept the extra burden of maintaining a separate GPU version of an application code. However, despite being considered ready for production, JURASSIC remains to be a research code under continuous development, i. e., it should be possible to add new developments without too large programming efforts at any time. Therefore, a single source policy has been pursued as much as possible. Having a single source also provides that practical advantage to compile and link a CPU and a GPU version inside the same executable.

Retaining a single source is a driving force for directives-based GPU programming models such as OpenACC, where CPU codes are converted to GPU codes by code annotations, comparable to OpenMP pragmas for CPUs. On the one hand, in order to harvest the best performance and to acquire maximum control over the hardware, CUDA kernels are considered mandatory. On the other hand, the coding complexity outlined above (kernels, drivers, launch parameters, grid stride loops, GPU pointers,





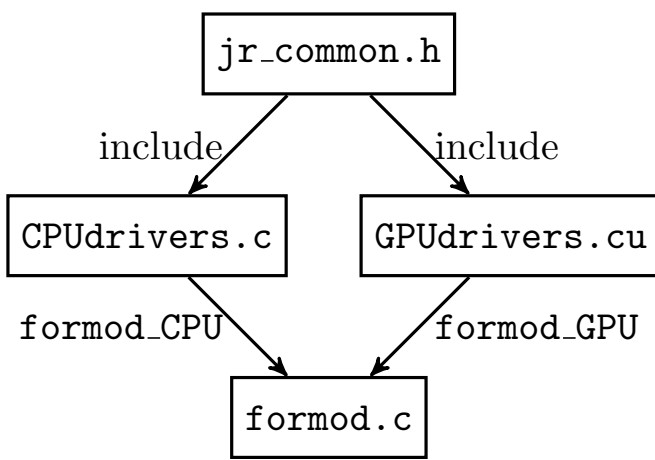

**Figure 4.** Common source code approach for CPUs and GPUs. The complete functionality of the JURASSIC forward model is provided via a joint header file included by both, the specific CPU and GPU drivers.

memory transfers) should remain hidden to some extent for some developers of the code, which may be domain scientists or students that are not familiar with all peculiarities of GPU programming. This poses a challenge for enforcing a single source policy.

For these reasons, the GPU-enabled version of JURASSIC is structured as follows: all functionality related to ray tracing and radiative transfer that should run on both, CPU and GPU, is defined in inline functions in a common header file `jr_common.h`. This header is included into the architecture-specific implementations `CPUdrivers.c` and `GPUdrivers.cu`. The implementation `CPUdrivers.c` contains the forward model to compute the radiative transfer based on the EGA method using OpenMP-parallelism over ray paths and instrument channels. The implementation `GPUdrivers.cu` is a CUDA source

file with CUDA kernels and corresponding drivers, which offers the same functionality as its CPU counterpart. Inside the CUDA kernels, the same functions are defined in `jr_common.h` as inside the loop bodies of `CPUdrivers.c`, see Fig. 4. With this approach, the user can decide at runtime whether to run the CPU or the GPU version of the model.

### 3.1.4   Code restructuring for improved performance

Using profile-guided analysis of execution runs of the JURASSIC reference code (Hoffmann, 2015), we identified that the most

critical component in radiative transfer calculations using the EGA method is the forward and backward interpolation between column densities $u$ and mean emissivities $\bar{\varepsilon}$ of the look-up tables. The emissivities $\bar{\varepsilon}$ are functions of pressure $p$, temperature $T$, and in particular they are monotonously rising functions of $u$. Furthermore, $\bar{\varepsilon}$ and $u$ carry an index for the specific trace gas and an index $g$ for the detector channel referred to by its wavenumber $\nu$. Therefore, interpolations on the emissivity look-up tables are conducted on 5-dimensional arrays of $\bar{\varepsilon}$ and $u$.

The look-up tables are densely sampled so that linear interpolation for all continuous quantities ($p$, $T$, $u$) can be assumed to be sufficiently accurate. Column densities $u$ are sampled on a logarithmic grid with increments of $12.2\,\%$. The starting value





of the grid can differ depending on $g$, $\nu$, $p$, and $T$. The emissivity curves for individual $(p, T)$ combinations feature up to 300 data points, i. e., a range of up to 15 orders of magnitude in $u$ is covered. However, for a given $u$ in order to interpolate between $\bar{\varepsilon}_i = \bar{\varepsilon}(u_i)$ and $\bar{\varepsilon}_{i+1} = \bar{\varepsilon}(u_{i+1})$ the index $i$ needs to be found such that $u_i \leq u < u_{i+1}$ on the logarithmic grid. A classical

bisection algorithm as implemented in the reference code of JURASSIC converges fast, but leads to a quasi-random memory access pattern in the look-up tables.

Depending on the number of emitters and detector channels, the total memory consumption of the look-up tables can become quite large. A typical configuration of sampling points leads to 3 MByte per gas and per instrument channel. Random memory accesses onto these memory sizes usually leads to an inefficient usage of the memory caches attached to the CPUs and GPUs.

The idea of caches is to reduce the memory access latency and potentially also the required bandwidth towards the main memory. A cache miss leads to a request for a cache line from the main memory, typically related to an access latency at least an order of magnitude larger than a read from cache. If the memory access pattern of an application shows a predominant data access structure, a large step towards compute and energy efficiency is to restructure the data layout such that cache misses are avoided as much as possible. Throughput-oriented architectures like GPUs work optimally when sufficiently many independent

tasks are kept in flight such that the device memory access latencies can be hidden behind computations on different tasks.

A random access to memory for reading a single float variable (4 Byte) may become particularly inefficient as always entire cache lines (32 Byte) are fetched from memory, i. e., only 12.5% of the available memory bandwidth is actually exploited. For exploiting the available bandwidth towards the memory optimally, we aimed to find a data layout that maximizes the use of a cache line.

The current JURASSIC reference implementation (Hoffmann, 2015) uses the following array-ordering to store the emissivity loop-up tables,

$$\bar{\varepsilon}_{g,\nu}(p,T,u) \longrightarrow \texttt{float eps[ig][inu][ip][iT][iu]}. \tag{24}$$

On GPUs, branch divergence is an important source of inefficiency. Therefore, the mapping of parallel tasks onto lanes (CUDA threads) and CUDA blocks has been chosen to avoid divergent execution as much as possible. For the computation of the EGA

kernels, lanes are mapped to the different detector channels $\nu$ and blocks are used to expose the parallelism over ray paths. Branch divergence may happen here only if a ray path gets optically thick in a given channel. However, this can be treated by masking the update to the transmittance $\tau$ on the corresponding lane.

Furthermore, coalesced loads are important to exploit the available GPU memory bandwidth. Although it seems counter-intuitive, the GPU version of JURASSIC therefore features a restructured data layout for the loop-up tables,

$$\bar{\varepsilon}_{g,\nu}(p,T,u) \longrightarrow \texttt{float eps[ig][ip][iT][iu][inu]}. \tag{25}$$

Compared to the data layout in Eq. (24), the index $\texttt{inu}$ has moved to the right. For each memory access to the look-up tables, we find that the four indices $\texttt{ig}$, $\texttt{ip}$, $\texttt{iT}$ and often also $\texttt{iu}$ are the same. Hence, vectorization over channels $\nu$ leads to a majority of coalesced loads and the best possible exploitation of the GPU memory bandwidth. Optimal performance has been found running with at least 32 channels (or even a multiple of that) as in this case, entire CUDA warps (groups of 32 lanes)

launch two coalesced load requests of 32 Byte each (Baumeister et al., 2017).

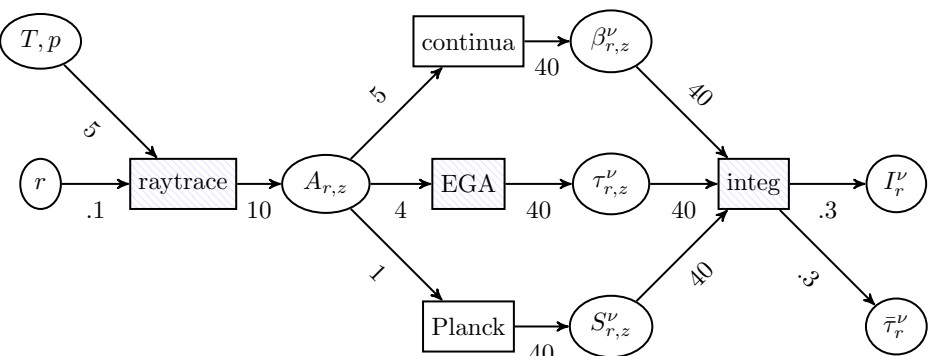

**Figure 5.** Data flow graph for the most important sub-kernels in JURASSIC. Data items are shown as ovals independent of whether they are stored in memory or exist only as intermediate results. Sub-kernels are depicted as rectangles. The arrow labels indicate the data sizes in units of kByte. The example refers to a nadir use case considering a single trace gas ($CO_2$) and 32 instrument channels. Figure adapted from Baumeister et al. (2017).

### 3.1.5 GPU register tuning

On CPUs, simultaneous multithreading, also known as hyper-threading, is achieved by assigning more than one thread to a core and by time-sharing of the execution time. This means that CPU threads can execute for a while until the operating system tells them to halt. Then, the thread context is stored and the context of the next thread to execute is loaded. GPUs can operate in
the same manner, however, storing and loading of the context can become a bottleneck as this means extra memory accesses. The best operating mode for GPUs is reached if the entire state of all blocks in flight can be kept inside the register file. Only then, context switches come at no extra cost. Consequently, GPU registers are a limited resource which we should monitor when tuning performance critical kernels. The NVIDIA C compiler can report the register usage and spill loads/stores of each CUDA kernel.
Figure 5 shows the data flow between kernels and sub-kernels for the complete forward model. Relevant data volumes are given as labels to the connector arrows in Fig. 5. As GPU kernels should not become too large in terms of register usage, an important implementation architecture choice is how to group sub-kernels into kernels. If two sub-kernels are grouped into the same GPU kernel, the data flow does not lead to stores to and consequent loads from GPU memory. This saves additional memory bandwidth and increases the potential maximum performance. The difficulty, however, is to balance between large
kernels which require many registers and, hence, reduce the number of blocks in flight and small kernels, which may cause additional memory traffic. We found that grouping the kernels `EGA`, `Planck`, `continua` and `integ` into a single, relatively large `fusion_kernel` delivers the best performance for nadir simulations (Baumeister et al., 2017). The `fusion_kernel` approach avoids additional memory traffic indicated by six connector arrows with 40 kByte per ray each.

     In the reference code of the forward model, the computation of continuum emissions requires many registers. The exact
number of registers depends on the combination of trace gases yielding continuum emissions in the given spectral range. A gas





**Table 1.** Register counts for the 16 possible combinations of switching on or off the $CO_2$, $H_2O$, $N_2$ and $O_2$ continua in the GPU `fusion_kernel` as reported by the NVIDIA compiler (`nvcc` v11.0.221) when compiling for an `sm_70` architecture such as NVIDIA V100.

| $CO_2$ | $H_2O$ | $N_2$ | $O_2$ | Number of Registers |
|------|------|-----|-----|---------------------|
| 0 | 0 | 0 | 0 | 77 |
| 0 | 0 | 0 | 1 | 82 |
| 0 | 0 | 1 | 0 | 84 |
| 0 | 0 | 1 | 1 | 84 |
| 0 | 1 | 0 | 0 | 87 |
| 0 | 1 | 0 | 1 | 92 |
| 0 | 1 | 1 | 0 | 94 |
| 0 | 1 | 1 | 1 | 94 |
| 1 | 0 | 0 | 0 | 78 |
| 1 | 0 | 0 | 1 | 82 |
| 1 | 0 | 1 | 0 | 84 |
| 1 | 0 | 1 | 1 | 84 |
| 1 | 1 | 0 | 0 | 87 |
| 1 | 1 | 0 | 1 | 92 |
| 1 | 1 | 1 | 0 | 94 |
| 1 | 1 | 1 | 1 | 94 |

continuum is considered relevant if any of the detector channels of the radiative transfer calculations falls into its predefined wavenumber window. JURASSIC implements the $CO_2$, $H_2O$, $N_2$ and $O_2$ continua, hence, there are 16 combinations of continuum emissions to be considered. In the spirit of multi-versioning, we use the C-preprocessor to generate all 16 combinations of the `continua`-function at compile time and jump to the optimal version with a `switch`-statement at runtime.

Table 1 shows the GPU register requirements of `fusion_kernel`. The maximal occupancy, i. e., the number of blocks or warps in flight, is limited by the total number of registers of the GPU's streaming multiprocessor (SM) divided by the register count of the kernel to execute. For example, the NVIDIA Tesla V100 GPU features $65,536$ registers in each of its $80$ SMs (NVIDIA, 2017). This defines a minimal degree of parallelism of $1760 - 2160$ ray paths (with 32 channels) in order to keep the GPU busy. Smaller register counts allow for a larger degree of parallelism and, potentially, a higher throughput.



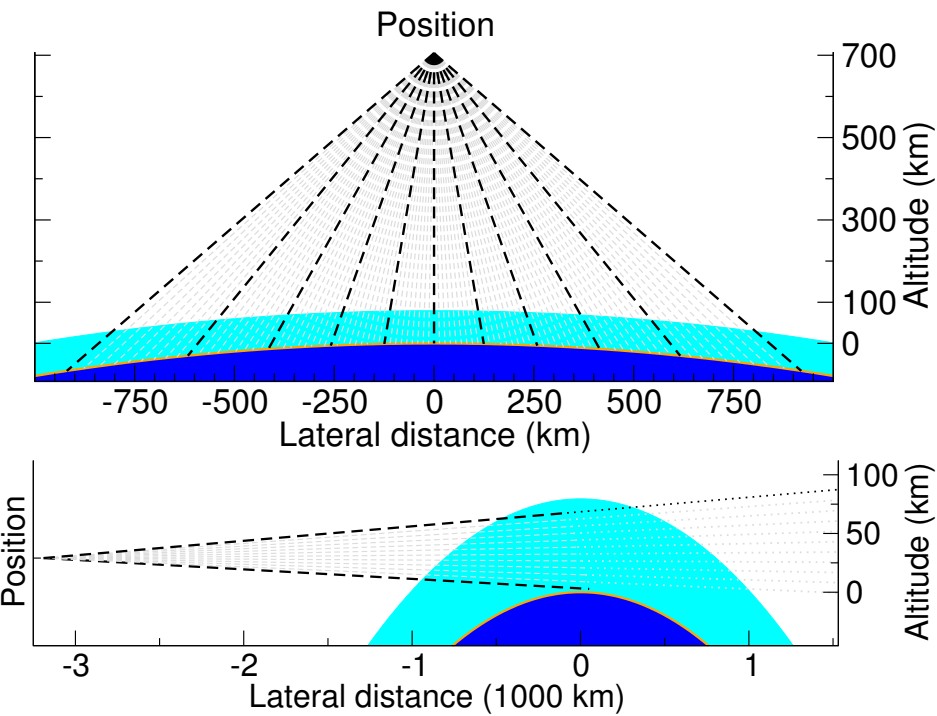

**Figure 6.** Nadir (top) and limb observation geometry (bottom) of satellite instruments.

## 3.2 Verification and performance analysis

### 3.2.1 Description of test case and environment

In the following sections, we discuss the verification and performance analysis of the JURASSIC-GPU code. All performance results reported here were obtained on the Jülich Wizzard for European Leadership Science (JUWELS) supercomputing system at the Jülich Supercomputing Centre, Germany (Krause, 2019). The JUWELS GPU nodes comprise a Dual Intel Xeon Gold 6148 CPU and four NVIDIA Tesla V100 GPUs (NVIDIA, 2017) of which one was used for benchmarking. A PCIe gen3 16x interface connects CPUs and GPUs. The CUDA runtime and compiler version are 10.1.105 while GCC version 8.3.0 was employed to compile the CPU code.

In the study of Baumeister et al. (2017), we analyzed the performance of the GPU implementation of JURASSIC for nadir applications. Here, we selected the limb geometry as a test case (c. f., Fig. 6). While in the nadir geometry ray paths typically comprise 160 path segments (assuming an upper height of the atmosphere at 80 km and default vertical sampling step size of 500 m), the limb geometry produces up to about 400 segments per ray path. Ray paths passing only through the stratosphere feature less segments than ray paths passing by close to the surface. Tropospheric ray paths are also subject to stronger refractive effects, bending them towards the Earth's surface. In the limb test case considered here, the number of segments along the ray paths varies between 122 and 393.





In this assessment, we aim for a rather extensive coverage of the mid-infrared spectral range. By means of line-by-line calculations with RFM, we prepared emissivity look-up tables for 27 trace gases with $1\,cm^{-1}$ spectral resolution for the range from 650 to $2450\,cm^{-1}$. Vertical profiles of pressure, temperature, and trace gas volume mixing ratios for mid-latitude atmospheric conditions were obtained from the climatological data set of Remedios et al. (2007).

    In order to verify the model, we continuously compared GPU and CPU calculations during the development and optimization
of JURASSIC-GPU. For the test case presented in this study, it was found that the GPU and CPU calculations are not providing bit-identical results. However, the relative differences between the calculated radiances from the GPU and CPU code remain very small ($\leq 10^{-5}$), which is orders of magnitudes smaller than typical accuracies of the EGA method itself ($\sim 1\%$) and considered suitable for most practical applications.

### 3.2.2    Performance signatures of emissivity look-up tables

It needs to be considered that many trace gases cover only limited wavebands throughout the mid-infrared spectrum (Fig. 7). In terms of the EGA calculations, typically between 7 and 18 look-up tables can be considered as being active throughout the spectrum (Fig. 8). The effect of the continuum emissions of $CO_2$ and $H_2O$ has been considered over the entire spectral range. The $N_2$ continuum was considered at 2120 to $2605\,cm^{-1}$ and the $O_2$ continuum was considered at 1360 to $1805\,cm^{-1}$.

    Figure 9 shows that the GPU runtime scales linearly with the number of active look-up tables per channel. Here, we compare
the GPU runtime to the number of active look-up tables averaged over a set of 32 channels that were processed together. For reference, we included also the actual number of active look-up tables for each single wavenumber as shown in Fig. 8. One can see the effect of the rolling average over 32 channels is smoothing out the steps.

    In order to quantify the correlation between the GPU runtime and the average number of active look-up tables, the data are presented as a scatter plot in Fig. 10. The scatter plot demonstrates mostly linear scaling between the number of active look-up
tables and the GPU runtime. A linear fit provides a quantitative measure of the GPU runtime with respect to the number of tables. The best fit to the data had a slope of $12.5\,\mu s$/table/ray and an offset of $16.7\,\mu s$/ray. While we would have expected that the calculations scale linearly with the number of active channels or trace gases considered in the EGA method, the offset can be largely attributed to the ray-tracing calculation, which need to be conducted once per ray, independent of the number of channels that are being considered.

### 3.2.3    The effect of cold caches

The average number of active look-up tables in the wavenumber interval from 650 to $2450\,cm^{-1}$ is 12.77. A subset of 32 channels from 1122 to $1153\,cm^{-1}$ matches this mean value well (with $409/32 = 12.78$) and has, therefore, been selected for a more thorough scaling analysis. The performance model generated in the previous section ($16.4\,\mu s + 11.3\,\mu s$ per look-up table) predicts a GPU runtime of approximately $161\,\mu s$/ray or a corresponding performance of 6.2 kray/s for this case while the actual
measured timing, $2.5\,s$ for 16,896 ray paths, accounts for a performance of 6.76 kray/s. This deviation is within the variance of the simple linear model.





**Figure 7.** Reference spectra for mid-latitude atmospheric conditions at $650$ to $2450\,\mathrm{cm}^{-1}$ and tangent heights from 5 to $50\,\mathrm{km}$. Shown are (a) total radiances considering the contributions of up to 27 trace gases as well as (b-f) atmospheres composed of only a single emitter.

As pointed out in Sect. 3.1.2, the emissivity look-up tables are allocated in the unified memory of the CPU and the GPU. The tables are first filled in by the CPU loading the corresponding data files from disk. After loading, no copy of the look-up tables resides in GPU memory, yet, which can be considered as some sort of cache in this context. For the GPU runtime measurements of the previous section, the actual computations have been repeated many times. The first iteration was considered as a warm-

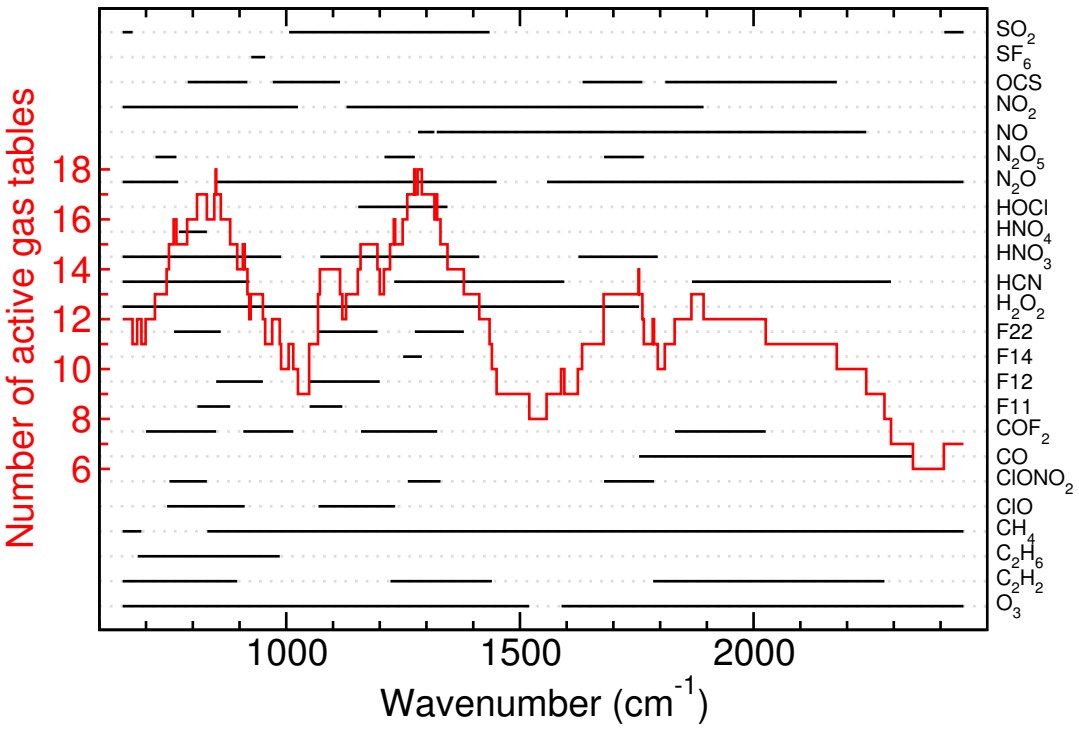

**Figure 8.** Spectral coverage of 24 selected gases between 650 and 2450 $cm^{-1}$. In addition to these gases, $CO_2$, $H_2O$ and $NH_3$ are considered over the entire spectrum. The total number of active gases varies between 6 and 18 as shown by the stair-type line.

up cycle and its timings are discarded. The performance model is therefore based on warm cache data only, i.e., there was already a copy of the EGA look-up tables being present in the GPU memory.

In order to understand the effect of cold caches, we repeated the measurements, but we were looking particularly onto the first timing result. With cold caches, the slope of the linear model for the runtime increased from 11.3 to 12.5 $\mu$s per table and per ray, which predicts a GPU runtime of approximately 176 $\mu$s/ray or correspondingly 5.7 kray/s for our test case. The offset only increased from 16.4 to 16.7 $\mu$s/ray. The actual measured timing, 2.74 s, accounts for a performance of 6.17 kray/s. This is a notable reduction of the performance from warm to cold caches.

From this, we can deduct the runtime increase due to memory page misses in the GPU memory. Table 2 shows that the model and direct measurements agree well in estimating an increased runtime of 0.25 s due to the cold cache. This increase of time, however, is much larger than the bare data transfer time of about 75 ms, which results from data volume of 1.2 GByte of the tables divided by the nominal bandwidth of 16 GByte/s of the CPU-GPU interconnect. Besides the bare transfer time, the increase in runtime is largely due to contributions from latencies, as every page fault is treated one-by-one. In the best case, some improvements due to overlapping of data accesses and computations on different blocks might be expected, a major strength of high-throughput devices like GPUs. However, this effect does not seem to come into play in this test case.





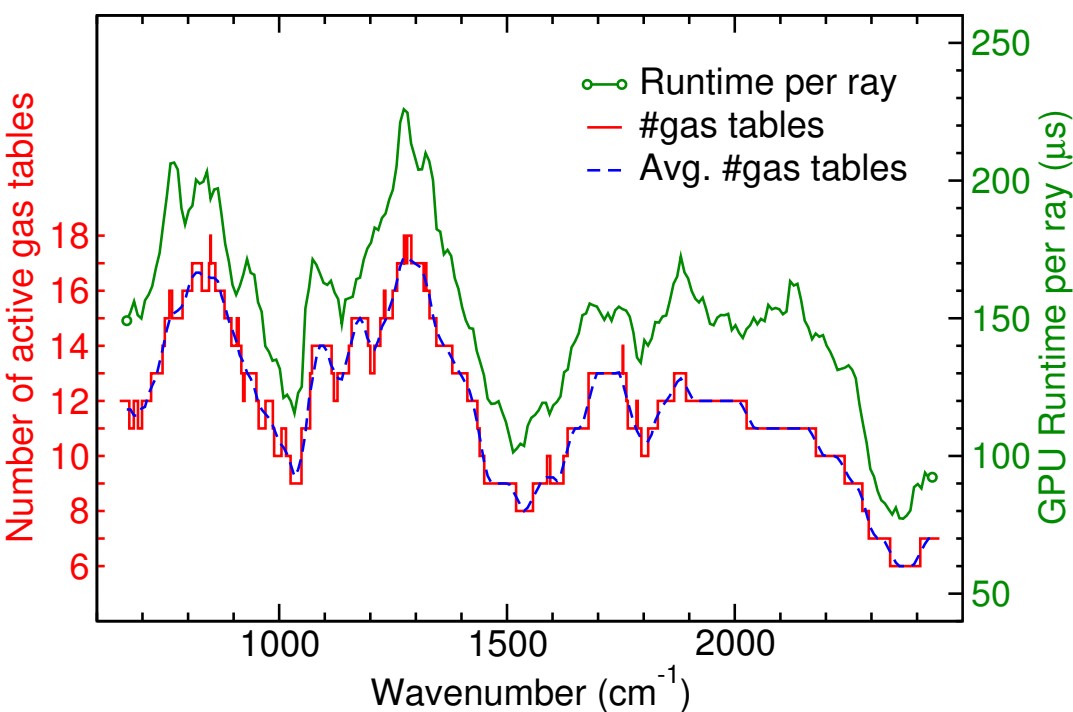

**Figure 9.** GPU runtime per ray as a function of wavenumber. Bundles of 32 wavenumbers from 650 to 2449 cm$^{-1}$ in steps of 8 cm$^{-1}$ have been benchmarked with up to 27 gases in the limb case. The averaged number of active look-up tables acts as a rolling average over 32 adjacent wavenumbers.

**Table 2.** GPU runtime measured and modelled for 16,896 limb ray paths with 12.78 of 27 look-up tables active. In the unified memory model, data is transferred into GPU memory on demand so a performance penalty is observed at first access (referred to as cold cache).

| | Case 1122 to 1153 cm$^{-1}$ | | Model with 12.78 tables | |
| --- | --- | --- | --- | --- |
| | $t$ (s) | $P$ (ray paths/s) | $t$ (s) | $P$ (ray paths/s) |
| Cache cold | 2.74 | 6,170 | 2.98 | 5,670 |
| Cache warm | 2.50 | 6,760 | 2.72 | 6,210 |
| Difference | 0.24 | -8.8 % | 0.26 | -8.7 % |

### 3.2.4 Workload scaling

In this section, we investigate the scaling behaviour of the GPU runtime with respect to the number of ray paths and the number of instrument channels. The GPU runtime results regarding the scaling with respect to the number of ray paths in Fig. 11 show two regimes: In the regime of small workloads (less than 1000 ray paths), overhead times of the order of 0.1 to 0.2 seconds become visible. For 32 channels and numbers of ray paths larger than 2k, the minimum amount of ray parallelism



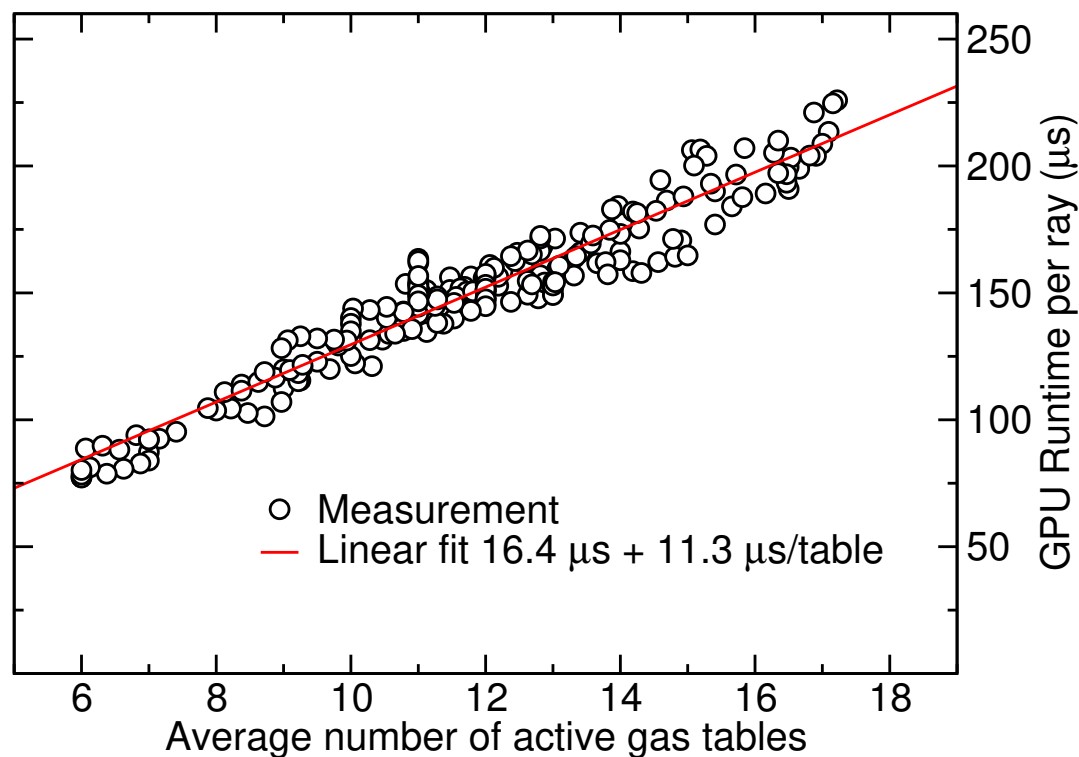

**Figure 10.** Linear scaling model to estimate GPU runtime: The GPU runtime per ray can be modelled as a linear function of the number of active look-up tables. For each ray, a V100 GPU processing 32 channels needs approximately $16.4\,\mu s + 11.3\,\mu s$ per active gas table. On the newer A100 GPU, a slope of $8.1\,\mu s$/table is found (not shown).

(c. f., Sect. 3.1.5), a linear behavior with a performance of up to 7.33 kray/s can be found. This is even about 8% higher than the best performance found earlier for 17 kray. We have to mention here that we reduced the maximum size of emissivity tables for this scaling analysis from 27 to 13 to host only the active gases.

The right subplot of Fig. 11 shows the scaling behavior with respect to the number of instrument channels. When using less than 32 channels, we can observe that the runtime is much higher than what we could expect from a linear function. This is strongly related to the GPU warp size of 32 lanes (also known as CUDA threads). For block sizes less than 32, branch divergence leads to inefficiencies and, hence, longer runtimes.

During earlier tuning efforts of the JURASSIC GPU version, much investigation was spent on radiative transfer calculations for the 4.3 and 15 micron wavebands of $CO_2$ and the nadir observation geometry (Baumeister et al., 2017). In that case, only look-up tables for $CO_2$ have been considered. In the nadir geometry, the total number of segments along the ray path is 160 for all ray paths and the best performance reported was 133 kray/s running on one NVIDIA Tesla P100 GPU.

The performance data for the limb case shown in Fig. 11 nominally translate into a maximum performance of 7.33 kray/s. However, the limb ray paths are longer (253.2 versus 160 segments) and we work with 13 rather than a single look-up table.

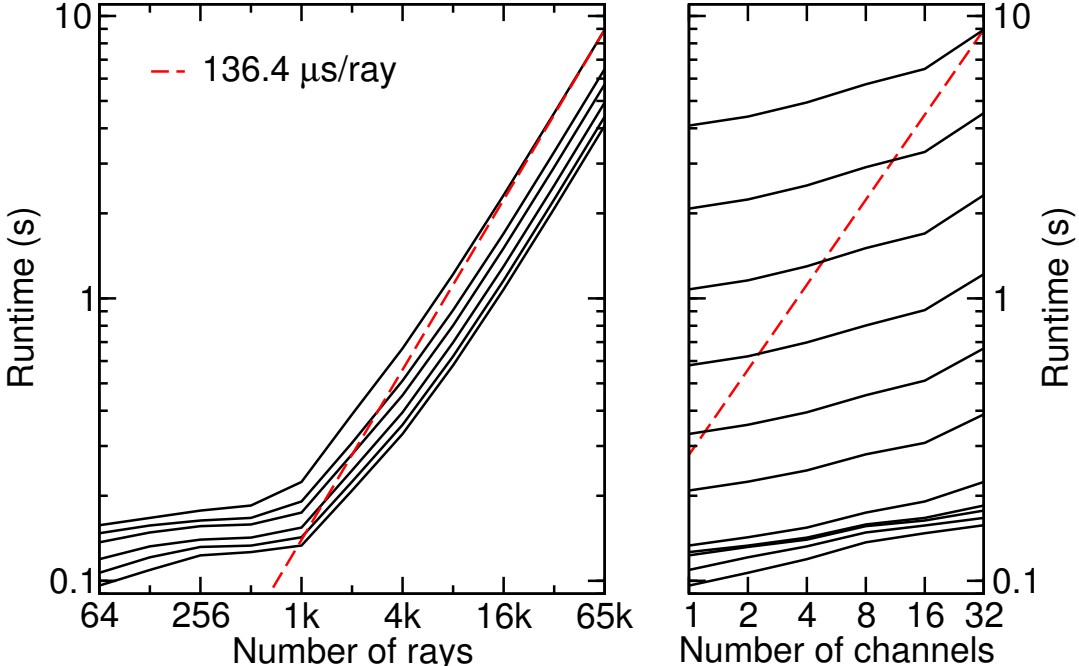

**Figure 11.** Scaling of the GPU runtime in the limb case. The left subplot shows the runtime as function of the number of ray paths for 1, 2, 4, 8, 16 and 32 channels. The right subplot shows the same data in a different projection: The different black lines refer to $2^6, 2^7, \ldots, 2^{16}$ ray paths from lowest to highest.

Accounting for that, the performance for the limb case on the V100 GPUs is about 11% higher than that of the nadir case on the P100 GPUs. The technology update from P100 to V100 increased the nominal GPU memory bandwidth by 25% (NVIDIA,

530 2016, 2017) but the nadir case using only $CO_2$ tables may also benefit stronger from caches compared to the limb case, where the 13 different look-up tables account for a total size of 1.2 GByte.

### 3.2.5 CPU versus GPU performance comparison

The code restructuring described above and by Baumeister et al. (2017) has been focused onto GPU performance, therefore, direct comparisons between tuned GPU and untuned CPU versions may be taken with a caveat. Nevertheless, we repeated the

535 scaling analysis of the previous section for the CPU and for the reference version (Hoffmann, 2015) (abbreviated as REF) in order to find a rough figure of merit.

The best performance results for the CPU version are achieved when running large workloads with 32 channels on two OpenMP threads per core. We find that the runtime is close to proportional to the number of ray paths over the entire range from 64 to 65k ray paths, see Fig. 12. Overhead times are negligible here. The right subplot indicates that running with

540 32 channels takes about twice the time of 16 channels. However, when we come to smaller channel numbers, overheads and inefficiencies become visible. In terms of overhead the channel-independent ray tracing needs to be mentioned here.





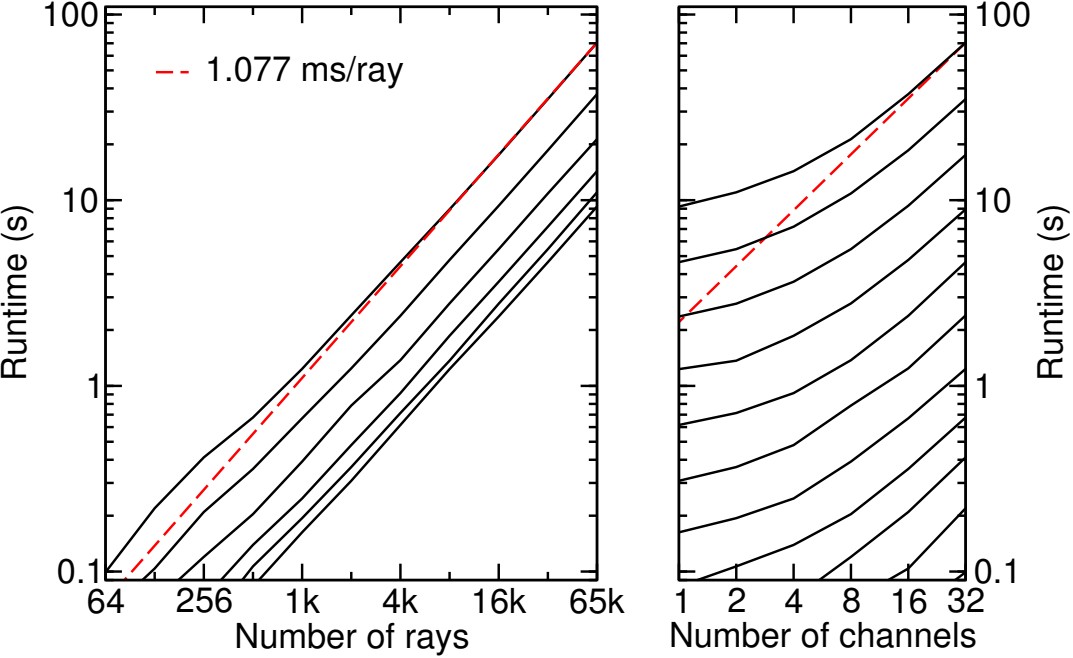

**Figure 12.** Scaling of the CPU runtime with respect to the numbers of ray paths. See caption of Fig. 11 for more detailed explanations. Mind the different scales.

Furthermore, inefficiencies in the EGA method appear as we cannot exploit the full width of the CPU vectorization (SIMD) with 1 or 2 channels.

The benchmarks of the reference version (REF) have been restricted to workloads of up to 4096 ray paths, as the code has not been optimized to handle larger workloads. However, Fig. 13 reflects its performance signature well enough. The runtime is proportional to the number of ray paths over the full range, i.e., no relevant overhead times are visible in the left subplot. The right subplot shows that the configurations with 16 and 32 channels take more than 2× and 4× the runtime, respectively, c.f., right subplot of Fig. 13. The black dot indicates the most efficient use case with 8 channels running on one OpenMP thread per core. One would choose this settings for a production calculation. The linear fit with 3.765 ms/ray also refers to the case of eight channels, so it must be scaled by 32/8 before comparing to CPU version results.

We extracted the best performance from GPU, CPU, and REF implementations and summarized them in Table 3. In order to serve for a direct comparison, the values for time-to-solution and energy-to-solution have been normalized to the same workload of $4 \times 65\,\mathrm{k}$ limb ray paths on 32 channels. As reported by peers about other code acceleration projects with GPUs, we can observe that the CPU version benefits from the rewriting efforts: Comparing the runtime of the CPU to that of the reference implementation shows a 14× faster time-to-solution and, equivalently, as both run on the CPU only, a 14× better energy-to-solution assuming that the CPU power intake is in first order approximation independent of the workload.



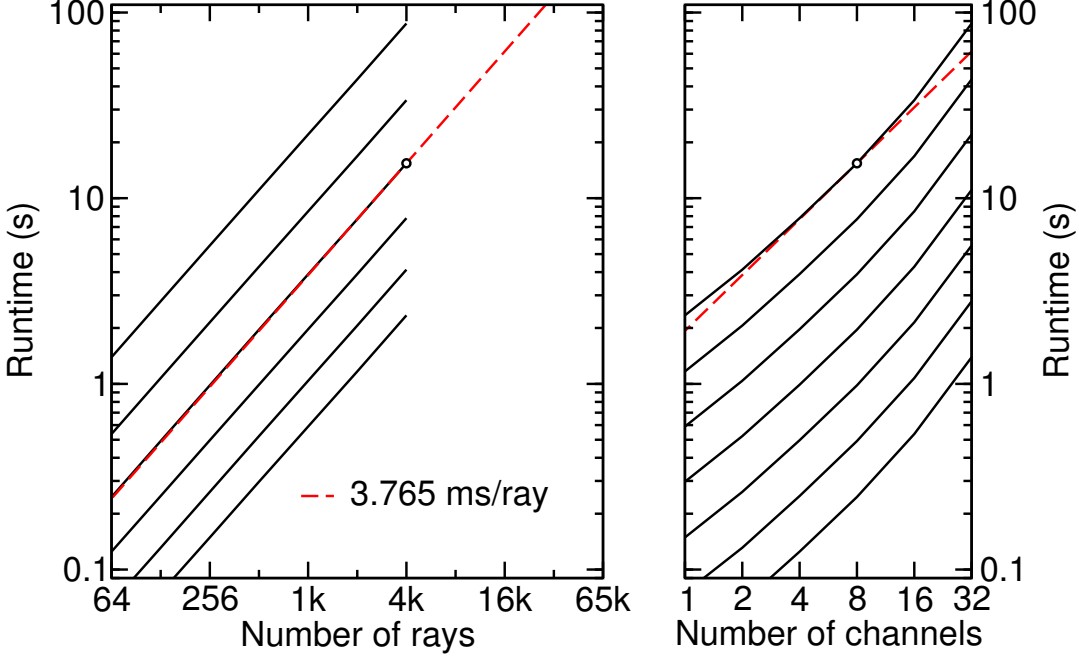

**Figure 13.** Scaling of the (REF) reference implementation runtime with the number of ray paths and channels. In contrast to Fig. 12, only up to 4 k ray paths have been benchmarked and the linear slope extrapolates to 65 k for 8 channels (black dot) since that configuration reached the best efficiency.

**Table 3.** Time-to-solution and estimated energy-to-solution comparison between GPU version, CPU version and the reference implementation (Hoffmann, 2015).

| Version | Processor | Runtime (s) | Energy (kJ) |
| --- | --- | --- | --- |
| REF | Intel Xeon Gold 6148 | 3950 | 1975 |
| CPU | Intel Xeon Gold 6148 | 282 | 141 |
| GPU | NVIDIA Tesla V100 | 9 | 15 |
| GPU | NVIDIA Tesla A100 | 6 | 11 |

A direct comparison of GPU runtimes to CPU runtimes is in most cases hardly meaningful as we need a CPU to operate a GPU. Therefore, we tried to estimate the power consumption of the compute node with and without GPUs active. We assume a thermal design power (TDP) envelope of 300 W for each of the four V100 GPUs (NVIDIA, 2017) and a TDP of 150 W for each socket of the Intel Xeon Gold 6148 CPU. Furthermore, the compute nodes are equipped with 192 GByte CPU memory such that we estimate the dual node to take 500 W without GPUs in operation and about 1.7 kW running a GPU accelerated application. Although these considerations do not include power for cooling, these energy-to-solution estimates allow to form meaningful energy efficiency ratios.





From Table 3, it becomes obvious that the GPU version is about $9\times$ more energy efficient than its CPU counterpart. Comparing to the reference version, the GPU version exhibits a more than $130\times$ better energy efficiency. However, this should be taken with a caveat as no particular CPU tuning has been undertaken. Nevertheless, we can roughly assign the improvement by the factor $14$ as the result of the rewriting and the factor $9$ as result of the GPU acceleration.

## 4   Conclusions

Numerical modeling of infrared radiative transfer on graphical processors (GPUs) can achieve considerably higher throughput compared to standard CPUs. This applies also for the case of the emissivity growth approximation (EGA), which allows us to effectively estimate band-averaged radiances and transmittances for a given state of the atmosphere, avoiding expensive line-by-line calculations.

In order to enable the GPU acceleration, including raytracing and the EGA method, major restructurings of the radiative transfer model JURASSIC have been necessary. Besides the goal of maximizing the GPU's throughput, the code base has been transformed to offer both, a GPU and a CPU version of the forward model, whereby the number of duplicate source code lines has been minimized facilitating better code maintenance. The GPU version of JURASSIC has been tuned to deliver outstanding performance for the nadir geometry in earlier work. In the nadir case, only $CO_2$ was considered as an emitter. In this work, we focused onto performance analyses for the limb geometry. Here, up to $18$ gases contributed to the simulated radiance in a given mid-infrared spectral region, leading to a much larger number of emissivity look-up tables that needs to be considered in the EGA method. Our scaling tests showed that the GPU runtime is composed of rather constant offset due to raytracing and a linearly scaling contribution due to the number of look-up tables being considered in the EGA calculations.

In order to find a figure of merit to evaluate the application porting and restructuring efforts for JURASSIC, we tried to assess the performance ratio of GPUs over CPUs. In terms of energy-to-solution we found the GPU version to be about nine times more energy efficient that its CPU counterpart. The CPU version, in turn, is about $14$ times faster than the reference implementation from which the porting project started. Although we developed further ideas for code optimization and performance tuning during the course of this study, the given achievements in terms of improved CPU performance and utilization of GPUs are considered an important step forward in order to prepare the JURASSIC radiative transfer model for large-scale data processing of upcoming satellite instruments.

*Code and data availability.*  The most recent version of the JURASSIC-GPU model is available at https://github.com/slcs-jsc/jurassic-gpu (last access: 2021 June 10). The release version 2.0 described in this study (Baumeister and Hoffmann, 2021) is accessible at https://doi.org/10.5281/zenodo.4923608. The code has been released under the terms and conditions of the GNU General Public License (GPL), version 3. The reference spectra used in this study and further test data are also included in the repository.



*Author contributions.* PFB and LH developed the concept for this study. PFB is the main developer of the GPU implementation and LH the main developer of the CPU reference implementation of JURASSIC. PFB conducted the verification tests and performance analyses of the JURASSIC-GPU code. Both authors made equal contributions to writing the manuscript.

*Competing interests.* The authors declare that no competing interests are present.

**Acknowledgments**

This work was made possible by efforts conducted in the framework of the POWER Acceleration and Design Center. We thank the Jülich Supercomputing Centre for providing access to the JUWELS Booster. We acknowledge the consultancy of Jiri Kraus (NVIDIA) and earlier contributions by Benedikt Rombach (IBM) and Thorsten Hater (JSC) to the software development and vivid discussions with Sabine Grießbach (JSC).



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
