# Peer review of "Fast Infrared Radiative Transfer Calculations Using Graphics Processing Units: JURASSIC-GPU v2.0"

_Geoscientific Model Development, 2021_

## Author Comment (AC1)

**Reply to review comments**

We thank the reviewers for the time and effort spent on the manuscript and for providing helpful comments. We considered all comments and hope that the revised draft properly addresses the open issues. Please find our point-by-point replies below (colored in blue). A revised manuscript with tracked changes has been uploaded.

**Reviewer #1**

**Summary**

The paper describes the JURASSIC radiative transfer model, which can be used to simulate the measurements of hyperspectral infrared instruments in both limb and nadir-viewing geometries. This is a fast, band-averaged RTM using the Emissivity Growth Approximation to approximate transmittances of complex paths along the line of sight.

Noting the expected large increase in the data volume from hyperspectral satellite instruments in the next few years, the second part of the paper describes some experiments implementing the code on a GPU system. However, this it outside my area of expertise so I limit my comments on this part to a few trivial, grammatical suggestions.

We would like to thank the reviewer for carefully reading and constructive comments and suggestions to improve the manuscript.

**Main Comments**

1) There should be a bit more acknowledgement and review of the approaches used in some of the other fast RTMs already used operationally. For example RTTOV which, like JURASSIC, is a band-averaged model, widely used in NWP applications, which has a different technique ('predictors') for evaluating transmittance of complex paths. Also the OSS model from AER, a high-speed monochromatic model which is a candidate for operational processing for future instruments.

We agree that more references to other fast RTMs should be provided, and the RTTOV and OSS models pointed out by the reviewer would be good examples. In the introduction, we added: 'For example, the Radiative Transfer for TOVS (RTTOV) model (Saunders et al., 1999, 2018) is a fast radiative transfer model for passive visible, infrared, and microwave downward-viewing satellite radiometers, spectrometers, and interferometers. In RTTOV, the layer optical depth for a specific gas and channel is parameterized in terms of predictors such as layer mean temperature, absorber amount, pressure, and viewing angle. The regression approach for the layer optical depths in RTTOV allows for particularly fast evaluation of the radiative transfer equation. Another example for rapid and accurate numerical modeling of band transmittances in radiative transfer is the optimal spectral sampling (OSS) method (Moncet et al., 2008). OSS extends the exponential sum fitting

of transmittances technique in that channel-average radiative transfer is obtained from a weighted sum of monochromatic calculations. OSS is well suited for remote sensing applications and radiance data assimilation in numerical weather prediction and a candidate for operational processing of future satellite missions.'

2) Clearly the number of path segments has a significant impact on computation. However it seems that the JURASSIC ray-tracing doesn't distinguish between the step size required for accurate ray-tracing (including, particularly, the integration of absorber amount within each segment), and the segmentation used for the subsequent EGA calculation and radiative transfer. In particular if a coarser spacing can be applied to the latter, there would be a significant saving in CPU time. Even so, I am surprised that the ray-tracing takes up so large a fraction of the computation time.

The reviewer is assuming correctly that the ray-tracing step size is used to establish the segmentation along the ray paths for the radiative transfer calculations. We tried to explain this more clearly, following the comment on L226 of the manuscript. As discussed near L226-237, it is also correctly assumed that the runtime for the radiative transfer calculations scales inversely with the step size. The raytracing step size is an important tuning parameter in terms of runtime and accuracy of the radiative transfer calculations. As outlined in the text, we generally apply a step size of 10 km along the path. However, the step size will be reduced as needed, to ensure the vertical component of each step will not exceeded 500 m. We empirically found that this scheme for selecting the step size works accurately and efficiently for a broad range a nadir and limb applications. In the past, we made experiments trying to use larger segments by calculating Curtis-Godson means along the path. However, this method was found to not yield any large benefits in terms of speed of the EGA calculations and was not pursued any further.

3) Regarding the 'continuum approximation' (L220 onwards). I believe a similar assumption is possible if the spectral features from the different absorbers are uncorrelated across the band. However, with hyperspectral instruments you are dealing with bands of less than 1 wavenumber, where there may be just a couple of individual lines, perhaps one from the target molecule and one from the an interfering molecule. So both the uncorrelated and continuum assumptions would seem to break down. Is this a fundamental limitation on the accuracy of JURASSIC compared, say, to other band models such as RTTOV, or do have you some proposed mechanism for dealing with such cases?

To clarify, we added: 'Some studies proposed to reduce the uncertainties associated with the continuum approximation by tabulating two-gas overlap terms (Marshall et al., 1994) or by introducing correction factors (Francis et al., 2006). These options might be interesting to further improve the accuracy of the approximated radiative transfer calculations, in particular for hyperspectral sounders measuring radiance in narrow spectral channels, but they have not been implemented in JURASSIC, so far. It is recommended that the errors associated with the band transmittance and continuum approximations are assessed by means of comparisons with line-by-line calculations for any specific case.' 4) Figure 1 and the associated text (L238) are very misleading. This suggests the Beer-Lambert Law is being used, ie complex path transmittance is simply the product of component homogeneous path segment transmittances. This is, of course, only true for monochromatic transmittances, not band-averaged transmittances as shown here. However, the error seems confined to just this part of the manuscript and is not, in fact, how JURASSIC actually evaluates path transmittances (as detailed in 2.6).

The numerical integration of the radiative transfer equation follows the scheme as outlined in the text and presented in Fig. 1. However, we realized, it might not be clear to the reader at this point, how the spectral mean segment emissivities are calculated. This is described only later in Sect. 2.6, as pointed out by the reviewer. To clarify, we added: 'It is essential to note that although the numerical integration scheme in Eq. (21) follows the Beer-Lambert law, the spectral mean segment emissivities  $\bar{\varepsilon}_k$  are determined by means of the EGA method (Sect. 2.6) in our model. The spectral mean emissivities from the EGA method along the path are different from and should not be confused with spectral mean emissivities that follow from treating the individual segments along the ray path as independent homogeneous gas cells.'

5) There are various references to using the 'bisection' method for interpolating the lookup tables (L256, L385). If I understand correctly this is basically an iterative approach, progressively dividing the tabulation axes by factors of two, and a significant CPU overhead. However, since your tabulation axes seem to be at regular intervals, surely you can evaluate your required interpolation nodes directly?

To clarify, we rewrote near L256: 'Direct index calculation is applied for regularly gridded data (temperature) and the bisection method (Press et al., 2002) is applied for irregularly gridded data (pressure, column density, and emissivities) in order to identify the interpolation nodes. There is potential to also exploit the regular structure of the column density grid in future versions.' The text near L385 describes the currently implemented way of interpolation of the column density.

**Minor comments**

L18: Suggestion: replace 'of about 4 and 15 um of wavelength' with 'from 4-15 um'

Changed to "from 4 to  $15\,\mu\text{m}$  of wavelength".

L19: Suggestion: start sentence 'Here, in the longwave part ...' to clarify that you regard the mid-IR as part of the longwave region

**Changed as suggested.**

L30: 'next to' ? Do you mean 'as well as' ? There should probably be a reference or two to go with this statement.

Changed as suggested. We added a reference to Collard and McNally (2009) on the use of IASI data for numerical weather prediction, whereas Clerbaux et al. (2009) discusses the use of IASI for monitoring the atmospheric composition.

L34: Perhaps also mention IASI-NG which will have a higher data rate than IASI, and the Chinese FengYun-4 GIIRS instrument already in orbit.

We like this suggestion and added references to IASI-NG and GIIRS.

L46: 'Massive amounts'. Perhaps 'a large number' sounds less excitable.

Changed as suggested.

L47: Suggestion '... provide a speedup factor of more ...'

Changed as suggested.

L60: 'Russell' (two 'L's) (also missing in several other places)

Fixed.

L69: 'Michelson' (one 's')

Fixed.

L71: Presumably 'for the Atmosphere' still contributes a part of the CRISTA-NF acronym

Fixed.

L101: It seems strange that JURASSIC should used a fixed value of Re. Can this really not be varied to match a particular viewing geometry (preferably without having to recompile the entire code)?

We rewrote to clarify: "For coordinate transformations between spherical and Cartesian coordinates, JURASSIC assumes the Earth to be spherical with a fixed mean radius  $R_e$  of 6367.421 km. The mean radius given here is considered in other radiative transfer models (e.g., Dudhia, 2017). It approximates the local radius of curvature with an accuracy of  $\pm 0.17$ %. Tests for the limb geometry showed corresponding differences in simulated radiances in the range of  $\pm 0.2$ % for tropospheric and  $\pm 0.1$ % for stratospheric tangent heights. The nadir geometry is not affected by applying the fixed mean radius of curvature." These errors are smaller than typical errors of the EGA method and we think the code does not need to be changed at this point.

L115: Suggest '4–15' (in LaTeX, or en-dash) to indicate continues range, rather than '4...15' which implies a sequence.

Changed as suggested.

L158: Delete 'being' - superfluous.

Fixed.

L175: The O2 and N2 continua arise from 'Collision Induced Absorption' which, in my mind, is a very different process from the CO2 and H2O continua which are largely the accumulation of line wings. Nowadays, HITRAN has a completely new file category for CIA (including these O2 and N2 continua), in parallel with the line database and molecular

cross-sections.

We agree that the different physical processes of the continua were misrepresented in the manuscript. We revised this sentence. Please see reply to additional comment forwarded by the Editor at the end of this reply.

L183: emissivity, in this equation, should be defined either before or immediately afterwards. L187 is a bit too remote.

We rephrased the paragraph to provide the definition of emissivity more early.

L219: '...small, if...' the comma is superfluous (in English grammar if not German!)

Fixed (here and in other places).

L226: How do you establish the pressure and temperature associated with the homogeneous conditions?

We rewrote this paragraph, to make it more clear: 'For the numerical integration of the approximated radiative transfer equation, Eq. (14), the ray path is first divided into segments. The segments are defined by the data points along the ray path as provided by the raytracing algorithm. For each segment, homogeneous atmospheric conditions are assumed. Pressure, temperature, trace gas volume mixing ratios, and aerosol extinction coefficients of each segment are calculated by applying the trapezoidal rule on neighboring data points along the ray path. The spectral mean emissivity and the value of the Planck function of each segment are calculated from segment pressure, temperature, trace gas volume mixing ratios, and aerosol extinction coefficients, respectively.'

L245: It might be useful to mention what units 'u' is measured in. And there should be an integral before the 'ds', along the length of the 'cell'.

We added a sentence stating the units and added the integral in the definition of the column density.

L275: Reference to Eq(19) suggests that a 'Delta e' term might be included after all. I assumed (from L.221) that it was not.

In principle, a correlation term  $\Delta \epsilon$  could be included in the radiative transfer calculations based on the EGA method. However, JURASSIC uses the approximation  $\Delta \epsilon \approx 0$ , as outlined in lines 210 to 223.

L289: epsilon\_2 - you seem to have switched notation. Here I understand this to represent the emissivity of both path segments combined but in L282 it would be the emissivity of just segment#2

We intended to use  $\varepsilon_2^*$  for the emissivity of the combined path, as pointed out by the reviewer. We fixed the notation errors in this paragraph.

L292-294: Is the appearance of the wavenumber nu significant or an oversight? In any case, I couldn't understand the point being made here - please try rewording it.

The wavenumber  $\nu$  was added intentionally, as we wanted to stress that an exact solution of the radiative transfer equation would require the detailed spectral shape of emissivities to match at the transition from segment #1 to segment #2. However, we realize we made a mistake as we used the symbol  $\bar{\varepsilon}$  for the spectral mean emissivity rather than  $\varepsilon$  for the spectrally resolved emissivity. We fixed this issue.

L303: Suggestion: 'computer' or 'computationally intensive' (unless 'compute intensive' is generally accepted usage - it's not really my field!).

We kept this as is, as 'compute intensive' seems to be a commonly used term (e.g., https://codata.org/rdm-glossary/compute-intensive/).

L377: 'monotonically' rather than 'monotonously' (which has now acquired a quite different meaning in English, as in 'tediously' or 'boringly').

**Fixed.**

L396 (and following). According to Wikipedia, kB and MB seem to be the recommended abbreviations.

Following the international standard IEC 80000-13 for units based on powers of 2, we replaced the units by KiB (kibibyte), MiB (mebibyte), and GiB (gibibyte) throughout the text.

L448: 'Wizard' (one 'z')

**Fixed.**

L447: (pedantically) 'fewer' rather than 'less' for discrete, countable items. Also L513, L519.

**Fixed.**

L473: Suggest 'from 2120 to 2605' rather than 'at 2120 to 2605', and again later in the same line.

**Fixed.**

L489 (and subsequently) I'm not keen on the introduction of a new unit, the 'kray'. I'd prefer  $6.2 \times 10^3$  rays/s.

**We replaced this as suggested.**

L493: 'After loading ... this context'. Wording could be improved ('yet' can mean both 'not yet' and 'although', which makes the overall meaning ambiguous).

We rephrased this sentence.

L503: 'deduct' - do you mean 'deduce', or did you really mean 'deduct' as in 'subtract' ? Changed to 'deduce'. L514: '2k' expand to 2000 (as with 1000 in previous line).

Fixed.

L528: 'performance ... 11% higher' Does this mean that the required time is 11% longer or shorter? 'higher' could be interpreted either way.

We replaced 'performance' by 'performance in terms of ray paths per second', to clarify.

L573: '... restructuring ... has ...' (singular)

Fixed.

```
L575: '... both a GPU ...' (remove superfluous comma after 'both')
```

Fixed.

**Reviewer #2**

The authors present a detailed manuscript regarding the implementation or adaptation of the JURASSIC (non-scattering) radiative transfer code/model for use with graphics processing units.

The first few sections give a short, but rather dense introduction into the method of emissivity growth approximation, a technique originally used for limb-type observation geometries, but also compatible with nadir-type observations. Experts in that field will probably have no trouble following Section 2, however outsiders might find that section a little jarring. There are some minor (apparent) inconsistencies in notation, but it is clear the focus of the paper is Section 3, with Section 2 being more of a recap - the authors provide enough references for further reading on the EGA method.

From Section 3 onwards, the manuscript delves into the challenges of transferring the JURASSIC implementation onto GPU architecture. This portion (which is also the central part of the publication) highlights some very interesting aspects of GPU implementations of an established code base, which itself is an interesting read to anyone who is looking to attempt a similar task (even if it is a different forward model). Towards the end, the authors make fitting comparisons of their GPU implementation compared to the CPU-only code, and then finally conclude with some easy-to-grasp numbers about the overall performance gain and an estimate on power consumption.

The whole manuscript was generally pleasant to read and offers interesting insight into the difficultions and considerations of writing GPU-accelerated scientific code. While it stays focused on the chosen algorithm itself, there are definitely some take-aways for other algorithms.

We would like to thank the reviewer for the encouraging feedback. We hope that, following

comments and suggestions provided by both reviewers, the recap on radiative transfer and the EGA method in Sect. 2 became more clear in the revised manuscript.

Minor suggestions:

General:

There seems to be no mention of linearization of the JURASSIC model for use in inversions. I suspect there is a straightforward way of calculating various Jacobians. I understand the paper focuses on the forward model aspect exclusively, however it would be nice to have it mentioned somewhere in the text (even if it is an acknowledgement that linearization has not been investigated).

We slightly rewrote the conclusions of the paper and pointed out that analytic Jacobians (linearization of the model) would be a promising idea for further increasing the performance of JURASSIC in data assimilation and retrieval applications in future work.

The Greek "mu" when used in units should be replaced by the 'upmu' (or equivalent) symbol.

Fixed as suggested.

L155: (Very minor) "T" in the Planck function is not explained in text.

Fixed.

L245: The equation should probably either say "du = qp / (kT) ds" or have an integral wrapped around the right-hand side.

Fixed (by inserting the integral).

L250++: Indices i,j,k are a little confusing as they're not explicitly written down.

We added the definition of the indices.

L275 and following: The emissivity notation (p,T,u) changed to (u,p,T) and then further (p, T, p, T, u) further down. This is a little confusing as well and would benefit from some more explicit explanation.

We revisited the text to make sure a consistent notation (p, T, u) is used. We rewrote the paragraph and fixed another issue with the notation to make it more clear. Please see reply to comments on L289 and L292-294 by Reviewer #1.

L350: (comment) I'm somewhat surprised that the emissivity look-up tables are not manually copied into GPU memory; I might be missing something, but it seems like the look-up tables would be one constant array (per species) and could be kept in memory for any given wavenumber window of the user's choice. In Section 3.2.3, the authors investigate cold/warm caches, so there must be a reason as to why this was not done - but if feels like a somewhat obvious choice to load the tables into GPU memory at the very start of the algorithm.

As pointed out in this paragraph, NVIDIA's unified memory approach was applied to automatically conduct data transfers of the look-up tables from CPU to GPU memory. The main reason for choosing the unified memory approach was that it is easier to implement, as no explicit data transfers need to be specified. Automatic data transfer of the look-up tables was not expected to be a performance critical part of the code, as the look-up tables need to be transferred only once at the begin of the simulations. The assessment on cold and warm caches confirms that using the unified memory approach for the look-up tables is indeed not performance critical. It certainly is correct that a manual triggering of the data transfer could remove some of the latency.

**Additional feedback (forwarded by editor)**

Continuum may also be collision-induced absorption, usually (as in Hitran) the spectral line features that are not explicitly modeled are gathered under some pure cross-section profile. It is anyways not good to attribute continuum to purely spectral lines. There is a need to either obfuscate (i.e., the exact details are not really important for the discussion, just as for line mixing, so just don't give any specific details and leave the reader to look it up themselves) or to specify the sentiment around line 174 on page 7.

We agree that the different physical processes of the continua were misrepresented in the manuscript. We followed the suggestion and obfuscated the details by shortening the sentence. Further details are provided in the references.

**References**

- Clerbaux, C., Boynard, A., Clarisse, L., George, M., Hadji-Lazaro, J., Herbin, H., Hurtmans, D., Pommier, M., Razavi, A., Turquety, S., Wespes, C., and Coheur, P.-F.: Monitoring of atmospheric composition using the thermal infrared IASI/MetOp sounder, Atmos. Chem. Phys., 9, 6041–6054, doi: 10.5194/acp-9-6041-2009, 2009.
- Collard, A. D. and McNally, A. P.: The assimilation of Infrared Atmospheric Sounding Interferometer radiances at ECMWF, Quart. J. Roy. Meteorol. Soc., 135, 1044–1058, doi: 10.1002/qj.410, 2009.
- Dudhia, A.: The Reference Forward Model (RFM), J. Quant. Spectrosc. Radiat. Transfer, 186, 243–253, doi: 10.1016/j.jqsrt.2016.06.018, 2017.
- Francis, G. L., Edwards, D. P., Lambert, A., Halvorson, C. M., Lee-Taylor, J. M., and Gille, J. C.: Forward modeling and radiative transfer for the NASA EOS-Aura High Resolution Dynamics Limb Sounder (HIRDLS) instrument, J. Geophys. Res., 111, doi: 10.1029/2005JD006270, 2006.

- Marshall, B. T., Gordley, L. L., and Chu, D. A.: BANDPAK: Algorithms for Modeling Broadband Transmission and Radiance, J. Quant. Spectrosc. Radiat. Transfer, 52, 581– 599, doi: 10.1016/0022-4073(94)90026-4, 1994.
- Moncet, J.-L., Uymin, G., Lipton, A. E., and Snell, H. E.: Infrared Radiance Modeling by Optimal Spectral Sampling, J. Atmos. Sci., 65, 3917 – 3934, doi: 10.1175/2008JAS2711.1, 2008.
- Press, W. H., Teukolsky, S. A., Vetterling, W. T., and Flannery, B. P.: Numerical Recipes in C, The Art of Scientific Computing, vol. 1, Cambridge University Press, 2. edn., 2002.
- Saunders, R., Matricardi, M., and Brunel, P.: An improved fast radiative transfer model for assimilation of satellite radiance observations, Quart. J. Roy. Meteorol. Soc., 125, 1407–1425, doi: 10.1002/qj.1999.49712555615, 1999.
- Saunders, R., Hocking, J., Turner, E., Rayer, P., Rundle, D., Brunel, P., Vidot, J., Roquet, P., Matricardi, M., Geer, A., Bormann, N., and Lupu, C.: An update on the RTTOV fast radiative transfer model (currently at version 12), Geosci. Model Dev., 11, 2717–2737, doi: 10.5194/gmd-11-2717-2018, 2018.